# BI-CRITERIA METRIC DISTORTION

**Kiarash Banishashem**[1]**, Diptarka Chakraborty**[2]**, Shayan Chashm Jahan**[1]**, Iman Gholami**[1]**,**
**MohammadTaghi Hajiaghayi**[1]**, Mohammad Mahdavi**[1]**, Max Springer**[3]
[1]Department of Computer Science, University of Maryland, College Park, MD 20742, USA
[2]School of Computing, National University of Singapore, Singapore
[3]Department of Computer Science, Princeton University, Princeton NJ 08540, USA
{kiarash, schjahan, igholami, hajiagha, mahdavi}@umd.edu
diptarka@nus.edu.sg maxspriger@princeton.edu

## ABSTRACT

Selecting representatives based on voters' preferences is a fundamental problem in social choice theory. While cardinal utility functions offer a detailed representation of preferences, voters often cannot precisely quantify their affinity towards a given candidate. As a result, modern voting systems rely on ordinal rankings to simplistically represent preference profiles. In quantifying the suboptimality of solutions due to the loss of information when using ordinal preferences, the metric distortion framework models voters and candidates as points in a metric space, with distortion bounding the efficiency loss. Prior works within this framework use the distance between a voter and a candidate in the underlying metric as the cost of selecting the candidate for the given voter, with a goal of minimizing the sum (utilitarian) or maximum (egalitarian) of costs across voters. For deterministic election mechanisms selecting a single winning candidate, the best possible distortion is known to be 3 for any metric, as established by Gkatzelis, Halpern, and Shah (FOCS'20). In contrast, for randomized mechanisms, distortions cannot be lower than 2.112, as shown by Charikar and Ramakrishnan (SODA'22), and there exists a mechanism with a distortion guarantee of 2.753, according to Charikar, Ramakrishnan, Wang, and Wu (SODA'24 Best Paper Award). Our work asks: can one obtain a better approximation compared to an optimal candidate by selecting a committee of $k$ candidates ($k \geq 1$), where the cost of a voter is defined to be its distance to the closest candidate in the committee? We affirmatively answer this question by introducing the concept of bi-criteria approximation within the metric distortion framework. In the line metric, it is possible to achieve optimal cost with only $O(1)$ candidates. In contrast, we also prove that in both the two-dimensional and tree metrics – which naturally generalize the line metric – achieving optimal cost is impossible unless all candidates are selected. These results apply to both utilitarian and egalitarian objectives. Our results establish a stark separation between the line metric and the 2D or tree metric in the context of the metric distortion problem.

## 1 INTRODUCTION

One of the fundamental challenges in the social choice theory is to elect representatives based on voters' preferences, ideally represented by cardinal utility functions that assign numerical values to each outcome. However, in most real-world scenarios, voters only provide ordinal information, such as preference orders among outcomes/candidates. This raises a natural question of how worse, if at all, a voting mechanism performs when given ordinal information rather than cardinal information. Procaccia & Rosenschein (2006) introduced the notion of *distortion* to measure such an efficiency loss – how different voting rules respond to the lack of cardinal information. Many practical voting scenarios can be formulated by considering both voters and candidates lying on a metric space (Enelow & Hinich, 1984). The distance to candidate locations determines voters' cardinal preferences for candidates – voters rank candidates based on ascending distance, with the closest candidate being the most preferable and the farthest candidate being the least preferable. The worst-case behavior of any ordinal preference order-based voting rule/mechanism is captured by the notion of

*metric distortion*, introduced by Anshelevich et al. (2018). A voting mechanism, without access to the actual distances among the set of voters and candidates, seeks to minimize a specific cost function, which depends on the distances. Distortion is defined with respect to this cost: for any voting mechanism $f$, it is the worst-case ratio, over all instances, between the cost of $f$'s solution and the optimal cost.

Given a single fixed candidate, the cost for a voter is defined as its distance from the given candidate. Then, the overall cost is set to be an objective that combines these values across all the voters. Depending on the specific context in the literature, the following two objectives have widely been considered: a utilitarian objective, which aims to minimize the total individual costs for all voters, and an egalitarian cost, which minimizes the maximum cost experienced by any voter. Different variants of the metric distortion problem under the above objective functions have received significant attention, e.g. Goel et al. (2017); Kempe (2020); Gkatzelis et al. (2020); Anagnostides et al. (2022); Kizilkaya & Kempe (2023a).

For the classical metric distortion problem, it is known that a distortion of 3 can be achieved for any metric. Further, it is widely known that the candidate chosen by any deterministic method cannot achieve a distortion factor of less than 3, even in a line metric. In other words, without knowing the exact distances, it is not possible to obtain a better than 3-approximation of the optimal cost. This leads to an intriguing question: Can we obtain a better approximation (distortion) by selecting more than one candidate? Specifically, assume that the algorithm is allowed to choose $k > 1$ candidates, and we set the cost for each voter to be its distance to the closest chosen candidate. Can we design an algorithm for which the overall cost (either utilitarian or egalitarian) is at most $\alpha$ times the cost of an optimal candidate for some $\alpha < 3$?

In this paper, we answer the above question affirmatively. We not only attain better distortion by allowing the selection of more than one candidate but, in fact, attain optimal cost with $O(1)$ candidates for the line metric. Additionally, we present several lower-bound constructions that demonstrate impossibility results for the line, tree, and 2D Euclidean metrics. Our findings establish matching upper and lower bounds for these cases. Furthermore, our results establish a separation between the line metric and the 2D Euclidean (and tree) metrics in the context of the metric distortion problem.

Our work introduces a *bi-criteria* perspective to metric distortion, which, to our knowledge, has not been considered before. The pursuit of improved *bi-criteria* approximation results for various classical optimization problems has already been well-investigated in the literature. Indeed, when the metric is known, and the goal is to pick $k$ candidates that minimize the overall cost across voters, the election problem becomes an instance of either the $k$-median or the $k$-center clustering (for the utilitarian and egalitarian objective, respectively). For these problems, numerous (constant-factor) approximation algorithms are known which select up to $O(k)$ centers (instead of $k$ centers), e.g. Feldman et al. (2007); Wei (2016); Alamdari & Shmoys (2018). Therefore, it is quite natural to explore a similar question in the metric distortion problem, where the underlying metric is not directly given.

Our work additionally extends the existing line of research on $k$-committee elections Faliszewski et al. (2017); Elkind et al. (2017); Caragiannis et al. (2022), which also select a committee of $k$ candidates and aim to minimize some loss function across all voters. The key distinction is that we consider a single candidate as our baseline for calculating distortion, while these works consider a baseline of $k$ candidates (see Section 1.2). Specifically, for the utilitarian cost we consider (i.e., the sum of distances of voters to their closest candidate in the committee), Caragiannis et al. (2022) show that the ratio of the cost for any deterministic method compared to the cost of an optimal committee can be unbounded in the worst case. This result fails to suggest a choice of candidates, as all possible choices are the same in the worst case. In contrast, we show that by using our baseline, one can make a meaningful distinction between different choices even though the underlying metric is unknown.

**Our contribution.** Our first result shows that when the underlying metric is a line, a committee of two candidates can achieve a 1-distortion of one under the *sum-cost* objective. Here, *1-distortion* refers to the worst-case ratio between the cost of the selected committee and the cost of an optimal single-winner candidate. For any voting rule $f$, its *distortion*, or more specifically, *1-distortion* is

$$\texttt{1-distortion}(f) := \sup_{\mathcal{E}} \sup_{d \triangleright \mathcal{E}} \frac{\texttt{cost}(f(\mathcal{E}))}{\texttt{OPT}}$$

where the cost function cost in the above definition could be either $\text{cost}_s$ or $\text{cost}_m$ depending on the context. In other words, the 1-distortion compares the cost of the mechanism to the cost of an optimal candidate in the worst case.

In fact, we will prove a stronger result by showing that it is possible to output a list of two candidates that always contains an optimal one. Formally, we prove the following theorem.

**Theorem 1.** An algorithm for the 2-committee utilitarian election on the line metric exists that guarantees the choice of an optimum candidate in the elected committee. Consequently, the 1-distortion of the algorithm is 1.

We further show that when the metric is not a line, one can obtain a distortion of $1 + \frac{2}{m-1}$.

**Theorem 2.** There exists an algorithm for the $(m-1)$-committee utilitarian election on the general metric that obtains a 1-distortion of, at most, $1 + 2/(m-1)$. Additionally, no algorithm can obtain a 1-distortion better than $1 + 2/(m-1)$ when choosing $m-1$ candidates, even if the metric space is 2D Euclidean or tree metric.

We further study the *max-cost* objective, showing that one can obtain 1-distortions of 1, 1.5, and 2 using sets of size four, three, and two, respectively.

**Theorem 3.** For any $k \in \{2,3,4\}$, there exists an algorithm for the $k$-committee egalitarian election on the line metric, which obtains a 1-distortion of at most $3 - k/2$. Furthermore, no algorithm can obtain a 1-distortion better than $3 - k/2$.

For the *max-cost* objective, even though there exists an algorithm that selects one candidate with a distortion of 3, we show that no algorithm can achieve a 1-distortion better than 3, even when choosing a committee of $m-1$ candidates.

**Theorem 4.** There is no algorithm for the $(m-1)$-committee egalitarian election on 2D Euclidean metric such that it can obtain a 1-distortion better than 3.

It is important to note that the obtained lower bounds apply to deterministic algorithms.

Table 1: State-of-the-Art; blue bounds are results of this paper.

| Objective | Metric Space | Committee Size (Out of $m$) | Lower-Bound | Upper-Bound |
|---|---|---|---|---|
| Sum | 1D | $\geq 2$ | 1 | $1^{a}$ |
| | | 1 | $3^{b}$ | $3^{c}$ |
| | 2D / Tree Metric | $m-1$ | $1 + \frac{2}{m-1}$ | $1 + \frac{2}{m-1}$ |
| Max | 1D | $\geq 4$ | 1 | 1 |
| | | 3 | 1.5 | 1.5 |
| | | 2 | 2 | 2 |
| | | 1 | 3 | 3 |
| | 2D | $m-1$ | 3 | 3 |

[a] Implied by Anshelevich & Postl (2017).
[b] From Anshelevich et al. (2018).
[c] From Gkatzelis et al. (2020).

## 1.1 TECHNICAL OVERVIEW

**Upper bound results.** Our approach to the bi-criteria metric distortion problem builds upon the foundational result by Elkind & Faliszewski (2014), who demonstrated that the ordering of voters and candidates can be deduced from ordinal preferences in a line metric. We use this established ordering for our upper bound results, enabling us to develop algorithms that select small committees achieving optimal or near-optimal distortion relative to the single-winner benchmark. We employ a distinct method to prove this ordering, which facilitates extensions of our results to more complex

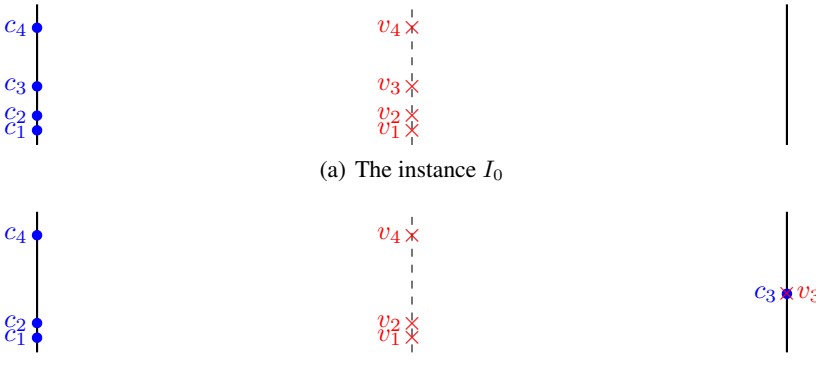

(a) The instance $I_0$

(b) $I_3$: Instance with voter $v_3$ and candidate $c_3$ moved

Figure 1: Figures of the lower bound instances. In (a), all candidates are located on the line $x = -\ell$, with voters with matching $y$-coordinates on the line $x = 0$. In (b), voter $v_3$ and candidate $c_3$ are moved to the line $x = \ell$, while keeping the same $y$-coordinate.

settings and yields improved outcomes in these generalized spaces. For the sake of completeness and to provide a clear foundation for understanding the remainder of our solution, we present a detailed exposition of this method in Section A.

Finally, to achieve the optimal solution for the sum objective with two candidates, we select candidates immediately to the left and right of any median voter. This approach follows from the result in Anshelevich & Postl (2017), with a detailed explanation provided in Section 3. In contrast, achieving a 1-distortion of 1 for the max objective requires a different strategy. We begin by identifying the leftmost and rightmost voters, whose distance is denoted by $D$. For the single-winner case, a lower bound of $D/2$ is known. Then, depending on whether we use 2, 3, or 4 candidates, we place them accordingly to achieve a 1-distortion of 2, 1.5, and 1, respectively.

**Lower bound results.** In Section 4, we provide various lower bounds for different settings. For the sum objective, we begin by presenting an example showing that any mechanism for choosing a committee of size $k$ cannot achieve bounded distortion, even when compared to an optimal choice of $\lceil \log(k+1) \rceil + 1$ candidates. This example relies on a binary tree structure with $k+1$ leaves, where each candidate corresponds to a leaf in the tree. The set of voters is the same as the candidates. Then, for any candidate, we construct an instance where the candidates whose lowest common ancestor with this candidate in the tree is a given vertex are in a close cluster. By selecting one representative for each cluster, we obtain a set of candidates of size $\lceil \log(k+1) \rceil + 1$, for which the total cost can be made arbitrarily small. In contrast, any solution excluding this candidate will incur a constant cost. Notably, these instances lie on the real line, demonstrating that the lower bound applies even for the line metric.

Next, we present examples in the 2D Euclidean metric space and tree metric space, where achieving a 1-distortion better than $1 + \frac{2}{m-1}$ for the sum objective is not possible even when choosing $m - 1$ candidates. To argue that, we consider a set of $m$ candidates and $m$ voters. Then, we create $m + 1$ instances $I_0, I_1, \cdots, I_m$. Each of them is distinct in terms of voters' and candidates' exact locations; however, they are indistinguishable in terms of voters' preference orderings (that are available as input to any voting mechanism). See Figure 1 to have an idea about the construction of the base instance $I_0$ and instances $I_j$ for each voter $j$. We show that not selecting any particular candidate would result in one of the above $m + 1$ election instances to attain a 1-distortion of at least $1 + \frac{2}{m-1}$.

This example highlights a drastic contrast between the line metric and higher-dimensional metrics. While a 1-distortion of one can be achieved on the line with just two candidates, this is not possible in higher dimensions, even when selecting all but one candidate. We also present a similar example for the max objective, demonstrating that achieving a 1-distortion better than 3 is impossible in the 2D Euclidean metric, even when selecting $m - 1$ candidates. Finally, we provide examples showing that our positive results for the 1-distortion of the max objective on the line are tight.

## 1.2 RELATED WORKS

The metric distortion framework has been central in analyzing single-winner voting. Gkatzelis et al. (2020) proved that any deterministic mechanism must have distortion at least 3, later simplified by Kizilkaya & Kempe (2023b). While it was conjectured that randomization could improve to 2, this was disproved by Charikar & Ramakrishnan (2022) and Pulyassary & Swamy (2021). Recently, Charikar et al. (2024b) achieved 2.753 with randomization. Separately, Anshelevich et al. (2024) showed that if threshold approval sets are known, distortion can be reduced to $1 + \sqrt{2}$.

In the $k$-committee election problem (the single-winner election being a special case with $k = 1$), the aim is to select $k$ candidates from a pool of $m$ candidates based on ordinal preferences provided by $n$ voters. For any mechanism $f$, its distortion is defined as the worst-case ratio (across all instances) of the cost of the solution produced by $f$ compared to the optimal cost. When the cost for a voter is considered as the sum of distances to all committee members, Goel et al. (2018) showed that the problem reduces to the single-winner election.

On the other hand, Caragiannis et al. (2022) considered a general cost function – each voter's cost is the distance to the $q$-th (for some integer $q \geq 1$) nearest committee member. They identified a trichotomy: For $q \leq k/3$, the distortion is unbounded; for $q \in (k/3, k/2]$, it is $\Theta(n)$; and for $q > k/2$, the problem reduces to the single-winner election. As an immediate corollary, for the 2-committee election, with each voter's cost being its distance to the nearest committee member (i.e., $q = 1$), we get a distortion of $\Theta(n)$. Further, with the same cost function, for the $k$-committee election when $k \geq 3$, the distortion is unbounded (even for the line metric). However, when the positions of candidates are known, for $k = m - 1$, Chen et al. (2020) demonstrated that single-vote rules achieve a distortion of 3 and provided a matching lower bound.

One of the most basic versions–where both voters and candidates are positioned on a real line– has already garnered significant attention in computational social choice theory. Strict preference profiles, with voters and candidates on a real line (also known as 1-D Euclidean), exhibit many intriguing properties, including being single-peaked and single-crossing (Black, 1948; Mirrlees, 1971; Escoffier et al., 2008). Given a preference profile and the order of voters, deciding whether it is 1-D Euclidean can be done in polynomial time (Elkind & Faliszewski, 2014). Furthermore, if the input preference order is consistent with 1-D Euclidean, Elkind & Faliszewski (2014) provides an efficient construction of a mapping realizing that. Very recently, Fotakis et al. (2024) studied the $k$-committee election problem on the 1-D Euclidean metric by allowing a few distance queries in addition to the voters' preference orders.

One closely related question to the problem of optimal candidate selection is the facility location problem Mahdian et al. (2006), where the goal is to place facilities at locations in a metric space to minimize the cost of serving agents. Unlike the candidate selection problem, where candidates are restricted to a fixed set, facilities in the facility location problem can be placed anywhere in the space Feldman et al. (2016). Another related concept is the Condorcet winning set: a set of candidates such that no other candidate is preferred by at least half the voters over every member of the set Elkind et al. (2015). The Condorcet dimension, defined as the minimum cardinality of a Condorcet winning set, is known to be at most logarithmic in the number of candidates, partially reaffirmed by Caragiannis et al. (2024). In the metric ranking framework, Lassota et al. (2024) demonstrated that the Condorcet dimension is at most three under the Manhattan or $\ell_\infty$ norms. Recently, Charikar et al. (2024a) showed that Condorcet sets of size six always exist.

## 2 PRELIMINARIES

**Metric space.** Let us consider a domain $\mathcal{X}$ and a distance function $d : \mathcal{X} \times \mathcal{X} \to \mathbb{R}$. We call $(\mathcal{X}, d)$ a *metric space* if the distance function $d$ satisfies the following properties:

- **Positive definite**: For all $x, y \in \mathcal{X}$, $d(x, y) \geq 0$, and $d(x, y) = 0$ iff $x = y$.
- **Symmetry**: For all $x, y \in \mathcal{X}$, $d(x, y) = d(y, x)$.
- **Triangle inequality**: For all $x, y, z \in \mathcal{X}$, $d(x, y) \leq d(x, z) + d(z, y)$.

In this paper, we consider

- **Line metric (1-D Euclidean metric)**: The domain is $\mathcal{X} = \mathbb{R}$, and for any two points $p, q \in \mathbb{R}$, their distance is $d(p, q) = |p - q|$.

- **2D Euclidean metric**: The domain is $\mathcal{X} = \mathbb{R}^2$, and for any two points $p = (p_x, p_y), q = (q_x, q_y) \in \mathbb{R}^2$, their distance is $d(p, q) = ||p - q||_2 := \sqrt{(p_x - q_x)^2 + (p_y - q_y)^2}$.

- **Tree Metric:** The domain is a set of vertices $V$, and there exists a weighted tree $T$ such that for any pair of vertices $u, v \in V$, the distance $d(u, v) = d_T(u, v)$, where $d_T(u, v)$ denotes the total weight of the path between $u$ and $v$ in $T$.

**Election.** An *election instance* $\mathcal{E} = (V, C, \succ)$ consists of a set $V = \{v_1, \ldots, v_n\}$ of $n$ voters and a set $C = \{c_1, \ldots, c_m\}$ of $m$ candidates. Each voter $v_i \in V$ has a linear order $\succ_i$ over the candidates, where $c_j \succ_i c_k$ indicates that voter $v_i$ prefers $c_j$ over $c_k$. We refer to $\succ_i$ as the *ordinal preference* of voter $v_i$. Furthermore, $\succ = \{\succ_1, \ldots, \succ_n\}$ is called the *preference profile* of the voters. Additionally, the $j$-th candidate in the ordinal preference of voter $v_i$ is denoted by $\succ_{i,j}$.

We consider the voters and candidates to lie in the same metric space $(\mathcal{X}, d)$. For ease of exposition, we extend the notion of distance $d$ to be defined directly on the set of voters and candidates instead of the points they occupy in the underlying space $\mathcal{X}$. We say a (distance) metric $d$ is *consistent* with an election instance $\mathcal{E} = (V, C, \succ)$, denoted as $d \triangleright \mathcal{E}$, when for any voter $v_i$, $c_j \succ_i c_k$ if $d(v_i, c_j) \leq d(v_i, c_k)$.

**Social cost.** Let us consider an election instance $\mathcal{E} = (V, C, \succ)$, and a distance metric $d \triangleright \mathcal{E}$. Let $I = (\mathcal{E}, d)$ denote the instance $\mathcal{E}$ with $d$ being its underlying distance metric. For any subset of candidates $S \subseteq C$, and a voter $v \in V$, we use $d(v, S)$ to denote the distance between the voter $v$ to its nearest neighbor in $S$, i.e., $d(v, S) := \min_{c \in S} d(v, c)$. In this paper, we focus on the following two *social costs*:

- **Sum-cost (Utilitarian objective)**: For any subset of candidates $S \subseteq C$, its *sum-cost*, denoted by $\mathrm{cost}_s(S, I)$ is defined as $\mathrm{cost}_s(S, I) := \sum_{v \in V} d(v, S)$.

- **Max-cost (Egalitarian objective)**: For any subset of candidates $S \subseteq C$, its *max-cost*, denoted by $\mathrm{cost}_m(S, I)$ is defined as $\mathrm{cost}_m(S, I) := \max_{v \in V} d(v, S)$.

When it is clear from the context, we drop $I$ and simply use $\mathrm{cost}_s(S)$ and $\mathrm{cost}_m(S)$.

**Voting rule and distortion.** A (deterministic) *voting rule* (also referred to as *mechanism*) $f$ is a function that maps an election instance $\mathcal{E}$ to a subset of candidates $S$. We use algorithms and mechanisms interchangeably throughout this paper. In this paper, we compare the cost of a voting rule that selects a $k$-sized committee with the cost of an optimal single candidate. We call a single candidate $c_{\mathrm{opt}}$ *optimal* if

$$\mathrm{cost}(c_{\mathrm{opt}}) = \min_{c \in C} \mathrm{cost}(c).$$

Throughout the paper, we use $\mathrm{OPT} = \mathrm{cost}(c_{\mathrm{opt}})$ to refer to the cost of an optimal single candidate. To capture how good a voting rule is in the worst case, the notion of distortion is used. For any voting rule $f$, its *distortion*, or more specifically, *1-distortion* is defined as

$$\mathtt{1\text{-}distortion}(f) := \sup_{\mathcal{E}} \sup_{d \triangleright \mathcal{E}} \frac{\mathrm{cost}(f(\mathcal{E}))}{\mathrm{OPT}}$$

where the cost function $\mathrm{cost}$ in the above definition could be either $\mathrm{cost}_s$ or $\mathrm{cost}_m$ depending on the context. In other words, the 1-distortion compares the cost of the mechanism to the cost of an optimal candidate in the worst case.

## 3 UPPER BOUNDS ON DISTORTION FACTOR

**Upper bounds for sum-cost.** We study the sum objective, where the cost of a candidate set is the sum of distances from voters to their closest selected candidate. On the line metric, using the orderings from Section A, we show that a committee of size two always contains an optimal candidate.

**Theorem 5.** There exists a voting rule for the 2-committee election for the sum-cost objective, on the line metric, such that the resulting committee includes an optimal candidate. Consequently, the 1-distortion of the voting rule is 1.

We proceed in two steps. First, we show that it is always possible to select three candidates such that one optimal candidate is guaranteed to be included. Second, we show that one of these three candidates can be safely discarded without ever removing that optimal candidate.

For the sum of distances, the median voter is the "balance point": moving a candidate toward the median decreases distance to the majority more than it increases it to the minority. Thus, an optimal candidate can be slid toward the median without increasing cost until it becomes one of the two candidates adjacent to the median voter (for an even number of voters, we fix either middle voter as the median). Note that the proofs of all lemmas and theorems appear in Appendix B.1.

**Lemma 6.** When considering the sum of distances objective and the candidates and voters are located on the real line, one of the candidates directly to the right or left of the median voter will be an optimal candidate.

Using only ordinal information, we order voters and candidates and discard candidates never closest to any voter. Let $c$ be the candidate closest to the median voter, and take the candidates immediately to the left and right of $c$ in the candidate ordering. By Lemma 6, the optimal candidate is the first candidate to the left or right of the median and must belong to this triple.

**Lemma 7.** There exists a voting rule for the 3-committee election on the line metric such that the resulting committee includes an optimal candidate.

To prove Theorem 5, let the three candidates in order on the line be $c_1, c_2, c_3$, and partition voters into three contiguous groups according to which of these is closest (this partition is determined by ordinal preferences). If more voters lie to the right of $c_2$ than to the left, replacing $c_3$ by $c_2$ cannot increase the sum of distances, so $c_2$ is at least as good as $c_3$; symmetrically, if the left side is heavier, $c_2$ is at least as good as $c_1$. Thus, one extreme is never strictly better than the middle candidate, and we can drop it while still retaining a 2-set containing an optimal candidate.

For a general metric, we show that choosing $k = m - 1$ candidates suffices to obtain 1-distortion $1 + \frac{2}{m-1}$. The rule looks only at each voter's top-ranked candidate and selects all but the least popular top choice. If the optimal single candidate is among these $m - 1$ winners, the distortion is 1; otherwise, it is the unique excluded candidate, so only voters ranking it first are affected. There are a few such voters by construction. Each can be rerouted, via a nearby voter whose favorite is selected, to an elected candidate with bounded extra distance, leading to total cost at most a factor $1 + \frac{2}{m-1}$ of the optimum.

**Theorem 8.** There exists a voting rule choosing $m - 1$ out of $m$ candidates achieving a 1-distortion of $1 + \frac{2}{m-1}$ for the sum-cost objective.

**Upper bounds for max-cost.** We now consider the max-cost objective, where the cost of a candidate set is the maximum distance of any voter to her closest selected candidate. Our benchmark is the best single candidate. We show that two, three, and four candidates suffice to guarantee distortions $2, 3/2$, and $1$, respectively.

Throughout this section, we use the leftmost and rightmost voters, denoted $v_l$ and $v_r$, identified using Section A. Since we only use each voter's top candidate, ties cause no ambiguity. Proof details are in Appendix B.2.

If $v_l$ and $v_r$ are far apart, no single candidate can be close to both: wherever the candidate is placed, at least one of $v_l$ or $v_r$ is at a distance of at least $d(v_l, v_r)/2$.

**Lemma 9.** If the distance between $v_l$ and $v_r$ is $D$ (i.e., $D = d(v_l, v_r)$), then the cost for the optimal single candidate OPT satisfies OPT $\geq \frac{D}{2}$.

Thus, the "span" $d(v_l, v_r)$ forces any single candidate to incur max-cost at least $D/2$.

From the orderings of Section A, we know how voters and candidates interleave on the line. When voters form a single block with no candidate strictly between $v_l$ and $v_r$, $c_l$ is the first candidate immediately to the left of all voters and $c_r$ the first to the right, and choosing $\{c_l, c_r\}$ is optimal.

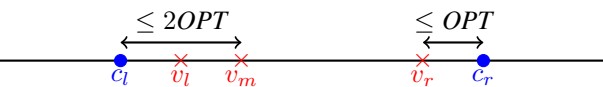

Figure 2: While the pair $\{c_l, c_r\}$ cannot achieve optimal 1-distortion, it guarantees a 1-distortion of 2.

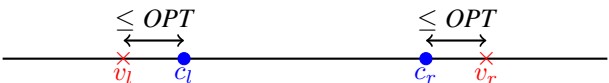

Figure 3: When $c_l$ and $c_r$ are between $v_l$ and $v_r$; $\{c_l, c_r\}$ is optimal.

**Lemma 10.** If there are no candidates placed between $v_l$ and $v_r$, a voting rule that selects $c_l$ and $c_r$, achieves a $1 - distortion$ of 1.

The nontrivial case is when at least one candidate lies between $v_l$ and $v_r$, which we assume from now on. Then any optimal single candidate $c_{\text{opt}}$ must also lie between $v_l$ and $v_r$; otherwise it is at least $d(v_l, v_r)$ away from one extreme voter.

Now $v_l$ and $v_r$ act as "anchors" for all voters. For any voter $v$ between them we have $d(v_l, v) + d(v, v_r) = D = d(v_l, v_r)$. By Lemma 9, $\text{OPT} \geq D/2$, so every voter is within distance $\text{OPT}$ of at least one of $v_l$ or $v_r$.

**Lemma 11.** If $d(v_l, v_r) = D$, then for every voter $v$, either $d(v, v_l) \leq \text{OPT}$ or $d(v, v_r) \leq \text{OPT}$.

This yields a 2-approximation:

**Theorem 12.** For any election $\mathcal{E}$ on the line metric, let $v_l$ and $v_r$ be the leftmost and rightmost voters, and $c_l$ and $c_r$ be the closest candidates to $v_l$ and $v_r$, respectively. Then, a voting rule $f$ that outputs the set $\{c_l, c_r\}$ satisfies $\texttt{1-distortion}(f) \leq 2$ with respect to the max-cost objective.

Indeed, $d(v_l, c_l) \leq \text{OPT}$ and $d(v_r, c_r) \leq \text{OPT}$ by definition, and Lemma 11 guarantees that every other voter $v$ is within distance $\text{OPT}$ of at least one of $v_l$ or $v_r$. Going via that extreme and then to $c_l$ or $c_r$ adds at most another $\text{OPT}$, so all voters are within $2\text{OPT}$ of $\{c_l, c_r\}$. The pair $\{c_l, c_r\}$ need not give distortion 1 when $c_l$ lies left of $v_l$ or $c_r$ right of $v_r$: some voters between $v_l$ and $v_r$ may then be far from both $c_l$ and $c_r$, even though $c_{\text{opt}}$ sits in the middle.

Conversely, if $c_l$ and $c_r$ both lie between $v_l$ and $v_r$ (Figure 3), then $\{c_l, c_r\}$ is optimal. In this case,

$$d(v, c_l) \leq \max(d(v, v_l), \text{OPT}), \quad d(v, c_r) \leq \max(d(v, v_r), \text{OPT}),$$

and by Lemma 11 each voter is within distance $\text{OPT}$ of $v_l$ or $v_r$, so

$$d(v, c_l) \leq OPT \quad \text{or} \quad d(v, c_r) \leq OPT.$$

To handle the general case, we "push" the selected candidates inward by adding more candidates. Adding one candidate just inside one extreme yields a 3/2-approximation; adding one near each side gives a distortion 1.

**Theorem 13.** For any election $\mathcal{E}$ on the line metric, there exists a voting rule to select three candidates, which achieves a 1-distortion of 3/2 with respect to the max-cost objective.

We start from $v_l$, $v_r$ and their closest candidates $c_l, c_r$. Using the candidate ordering, we add $c_r'$ immediately to the left of $c_r$ and consider $C = \{c_l, c_r, c_r'\}$. Any voter strictly outside the interval between $c_l$ and the rightmost candidate in $\{c_r, c_r'\}$ has her nearest candidate in $C$ by construction. For voters in the middle region, the distance between $c_l$ and that rightmost candidate is at most $3\text{OPT}$ (via $v_l$ and $c_{\text{opt}}$), so any voter between them is at a distance at most $3\text{OPT}/2$ from one of them.

**Theorem 14.** For any election $\mathcal{E}$ on the line metric, there exists a voting rule to select four candidates, which achieves a 1-distortion of 1 with respect to the max-cost objective.

For the 4-candidate rule, we symmetrize. From $c_l$ and $c_r$, let $c_l'$ be the next candidate to the right of $c_l$ and $c_r'$ the next to the left of $c_r$. Let $c_l^*$ be the one in $\{c_l, c_l'\}$ lying to the right of $v_l$, and $c_r^*$ the one

in $\{c_r, c_r'\}$ lying to the left of $v_r$. Consider $\{c_l, c_l', c_r, c_r'\}$. Every voter to the left of $c_l'$ or right of $c_r'$ has her closest candidate among these four. In the central part of the line, $c_l^*, c_r^*, c_{\text{opt}}$ lie between $v_l$ and $v_r$; since $v_l$ and $v_r$ are within distance $\texttt{OPT}$ of $c_{\text{opt}}$, we have $d(c_l^*, c_r^*) \leq 2\texttt{OPT}$, so any voter between them is within distance at most $\texttt{OPT}$ of one of them. Thus, every voter is within $\texttt{OPT}$ of the selected four, giving a distortion 1.

## 4  LOWER BOUNDS ON DISTORTION FACTOR

**Lower bounds for sum-cost.**  We now give lower bounds for distortion under the sum-cost objective. First, we extend the lower bound of Caragiannis et al. (2022) to show that bounded distortion is impossible for general $k$, even if we may select $\omega(k)$ candidates. The construction already works on the line metric. We encode each voter and candidate by an $\ell$-bit binary string (with $k = 2^\ell - 1$), define preferences so that a voter prefers candidates with longer common prefixes, and embed these strings on the line via a base-3 expansion where bits agreeing with a fixed "special" candidate receive large weight and later bits are downweighted by $\varepsilon$.

**Theorem 15.** For $k \geq 3$, there exists an instance for the $k$-committee election on the line (with respect to the sum-cost) where no voting rule choosing $k$ candidates can achieve a bounded distortion even when compared to an optimal choice of $\lceil \log_2(k+1) \rceil + 1$ candidates.

Any rule picking exactly $k = 2^\ell - 1$ candidates must omit some candidate $c_{i^*}$. We place voter $v_{i^*}$ at the same point as $c_{i^*}$ and arrange all other candidates so that every selected candidate is at a distance $\Omega(3^{-\ell})$ from $v_{i^*}$. Thus any $k$-sized committee omitting $c_{i^*}$ incurs a constant cost for $v_{i^*}$, independent of $\varepsilon$. In contrast, we build a "covering" committee of size $\ell + 1$: $c_{i^*}$ plus one candidate for each bit position where others first diverge from $b_{i^*}$. Since discrepancies beyond that position are downweighted by $\varepsilon$, every voter has a committee member at distance $O(\varepsilon)$, so by shrinking $\varepsilon$, the optimal cost can be made arbitrarily small. A $k$-sized committee excluding $c_{i^*}$ thus pays constant cost, whereas a $\Theta(\log k)$-sized committee has cost tending to 0, making the distortion unbounded.

Next, we construct instances in the plane and on tree metrics where no rule excluding at least one out of $m$ candidates can guarantee 1-distortion strictly less than $1 + \frac{2}{m-1}$.

**Theorem 16.** For any number of candidates $m$ and any $\varepsilon > 0$, there exist instances of the $(m-1)$-committee election problem in the 2D Euclidean metric for which no voting rule can guarantee a 1-distortion factor less than $1 + \frac{2}{m-1} - \varepsilon$.

**Theorem 17.** For any number of candidates $m$ and any $\varepsilon > 0$, there exist instances of the $(m-1)$-committee election problem with tree metrics for which no voting rule can guarantee a 1-distortion factor less than $1 + \frac{2}{m-1} - \varepsilon$.

The proofs of Theorems 16 and 17 share the same idea. We construct $m + 1$ metric instances $I_0, \ldots, I_m$ with identical candidate rankings but very different distances. In $I_0$, all candidates lie on one vertical line and all voters on a parallel line, so each voter $v_i$ is closest to the "matching" candidate $c_i$. For each $j \in [m]$, instance $I_j$ moves $v_j$ and $c_j$ far to the right, leaving all other points fixed; this preserves ordinal preferences, so any rule using only rankings must output the same $(m-1)$-subset on all instances.

In $I_j$, excluding $c_j$ greatly increases cost: $v_j$ is now at a distance of about $\ell$ from every selected candidate, whereas including only $c_j$ keeps all voters within a distance of roughly $\ell + 1$. Summing over voters, any $(m-1)$-committee omitting $c_j$ has cost on the order of $(m+1)\ell$, while the optimal cost is about $(m-1)(\ell+1)$. Thus

$$\frac{(m+1)\ell}{(m-1)(\ell+1)} \approx 1 + \frac{2}{m-1},$$

and by choosing $\ell$ large enough (as a function of $\varepsilon$) we make the deviation from this limit smaller than $\varepsilon$. The tree-metric version embeds the same "left vs. far-right" geometry into a tree (e.g., via long paths or a star) with the same effect.

**Lower bounds for the max-cost**  We now bound the distortion for the max-cost objective. In 2D Euclidean space, even when selecting $m - 1$ out of $m$ candidates, we cannot guarantee distortion less than 3, matching the distortion when selecting only one candidate. For the line metric, we then show that our algorithms that select $k = 2$ and $k = 3$ candidates are tight.

**Theorem 18.** Any deterministic algorithm for the $k$-committee election (with respect to the max-cost) that selects at most $k < m$ candidates out of $m$ candidates must have a 1-distortion of at least $3 - \varepsilon$ for any $\varepsilon > 0$.

We define $m + 1$ plane instances $I_0, \ldots, I_m$ with identical ordinal preferences. In $I_0$, all candidates lie on one vertical line, all voters on a parallel line, and each voter $v_i$ ranks $c_i$ first. For $I_j$, we push $c_j$ and $v_j$ far to the right so that $v_j$ is at a distance of about $3\ell$ from every candidate except $c_j$, while all other distances (and rankings) are unchanged. Any deterministic, ordinal rule must choose the same committee $C$ on all these instances. Since $k < m$, some candidate $c_j$ is not chosen; in $I_j$, the max-cost of $C$ is then driven by $v_j$, who sees all selected candidates at a distance of about $3\ell$, whereas the optimal solution picks $c_j$, keeping everyone within a distance of roughly $\ell + 1$. Hence

$$\frac{\text{cost}_m(C, I_j)}{\text{OPT}_m(I_j)} \approx \frac{3\ell}{\ell + 1} \approx 3,$$

and for large $\ell$ the distortion is at least $3 - \varepsilon$.

**Theorem 19.** Any deterministic algorithm for the 2-committee election (with respect to the max-cost) when voters and candidates are located on a line must have a 1-distortion of at least $2 - \varepsilon$ for any $\varepsilon > 0$.

For Theorem 19, we consider three voters and three candidates on a line with the cyclic preferences

$$v_1 : c_1 \succ c_2 \succ c_3, \quad v_2 : c_2 \succ c_1 \succ c_3, \quad v_3 : c_3 \succ c_2 \succ c_1.$$

We realize these rankings by three different metric instances on the line. In each instance, exactly one candidate is a "good center" with max distance at most 1 to every voter, and for each of the other two candidates, some voter is at distance $2 - \varepsilon$. Thus, in one instance, $c_1$ is uniquely optimal, in another $c_2$, and in the third $c_3$. Any deterministic 2-committee rule based only on rankings must output the same pair on all three instances. Whichever candidate it omits is uniquely optimal in one instance, where any 2-committee excluding it has max-cost at least $2 - \varepsilon$, whereas the optimal committee including it has cost 1.

**Theorem 20.** Any deterministic algorithm for the 3-committee election (with respect to the max-cost) when voters and candidates are located on a line must have a 1-distortion of at least $3/2 - \varepsilon$ for any $\varepsilon > 0$.

The proof of Theorem 20 uses four voters and four candidates on a line. We fix a symmetric preference profile where $v_1, v_2$ strongly favor $c_1, c_2$ and $v_3, v_4$ strongly favor $c_3, c_4$, and build four line instances $I_1, \ldots, I_4$ realizing these rankings but with different locations: in $I_i$, candidate $c_i$ is well-centered, at distance at most 2 from every voter, whereas voter $v_i$ is at distance at least $3 - 2\varepsilon$ from all other candidates. Thus, in $I_i$, the optimal max-cost is at most 2, achieved by including $c_i$, while any committee excluding $c_i$ must incur max-cost at least $3 - 2\varepsilon$ because of $v_i$. Any deterministic 3-committee rule must omit at least one candidate and cannot distinguish between the four instances by rankings alone, so for some $I_i$ it omits the uniquely good candidate. In that instance, its distortion is at least $\frac{3 - 2\varepsilon}{2} = \frac{3}{2} - \varepsilon$, showing that our upper bound of $3/2$ for the line metric with $k = 3$ is tight.

## 5 Conclusion and Future Work

In this paper, we initiate the study of the metric distortion problem under a bi-criteria approximation framework, namely whether one can match the cost of the optimal single candidate using only a fixed number of candidates. For the line metric, we show this is achievable: two candidates suffice for the utilitarian objective and four for the egalitarian objective, and for the latter we also give a smooth trade-off between committee size and approximation factor. We complement these results with matching lower bounds, establishing tightness on the line and showing that equally strong guarantees fail in richer metrics such as the 2D Euclidean plane or tree metrics.

Looking ahead, several directions remain. One is to fully characterize the metric spaces in which a constant-size committee can achieve optimal 1-distortion. Beyond this structural question, it is also natural to determine whether, in general metrics beyond the line, one can attain 1-distortion strictly better than the best achievable single-winner bound. Another direction is to investigate whether randomized mechanisms can improve guarantees in settings where deterministic mechanisms cannot attain optimal 1-distortion. One can also explore metric distortion and the $k$-committee election problem within this bi-criteria framework when only limited cardinal information is available.

## ACKNOWLEDGMENTS

This work is partially supported by DARPA expMath, ONR MURI 2024 award on Algorithms, Learning, and Game Theory, Army-Research Laboratory (ARL) grant W911NF2410052, NSF AF:Small grants 2218678, 2114269, 2347322. Diptarka Chakraborty was supported in part by an MoE AcRF Tier 1 grant (T1 251RES2303) and a Google South & South-East Asia Research Award.

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

## A    LINE METRIC ELECTION: THE ORDER OF CANDIDATES AND VOTERS

In this section, we present an algorithm to determine the order of candidates and voters for any line metric election instance $\mathcal{E} = (V, C, \succ)$ based on the voters' preference profiles. This step is essential for establishing the upper bounds discussed in Section 3. Moreover, this approach may prove useful for future research, as it offers a general method for obtaining the total order of candidates and voters, as explained below.

This algorithm focuses on a specific subset of candidates with properties useful for the purposes of this paper and potential future work on line metric distortion. This subset has the following property:

**Definition 21** (Core). In an election $\mathcal{E} = (V, C, \succ)$, a subset $A \subseteq C$ is called a *core* if for any voter $v_i \in V$, any candidate $c_j \in A$, and any candidate $c_k \in C \setminus A$, we have $c_j \succ_i c_k$.

Also, we are able to retrieve the order of voters based on this subset of candidates. Therefore, we define the following notation:

**Definition 22.** For a subset of candidates $A \subseteq C$, the order of voters *with respect to $A$* is a sequence $S$ of voters such that voter $v_i$ become before $v_j$ in $S$, if the preference order of candidates in $A$ by $v_i$ is lexicographically no greater than the preference order of candidates in $A$ by $v_j$, when $A$ is sorted according to some fixed order $S_A$.

Consequently, the main theorem of this section is the following:

**Theorem 23.** For an election $\mathcal{E} = (V, C, \succ)$ on the line metric such that $d \vartriangleright \mathcal{E}$, There exists an algorithm that identifies:

1. A *core* subset of candidates $C^*$;

2. The order of $C^*$ candidates, denoted as $S_C$; and

3. The order of voters *with respect* to $C^*$, denoted as $S_V$.

Based on Theorem 23, the algorithm introduced in this section determines the order only for a subset $C^*$, referred to as the *determined candidates*, rather than for all candidates. It expands $C^*$ iteratively and establishes the order of candidates within $C^*$. By the end of the algorithm, $C^*$ forms a core subset of candidates.

Additionally, for some voters with similar preferences, it may be impossible to distinguish whether they are on the right or left; thus, they are considered to be at the same point. These properties of the voter and candidate order retrieved by the algorithm are accounted for in the analysis and other sections of this paper. Overall, obtaining the order of voters and candidates with this level of accuracy remains significant for the intended purposes.

The algorithm consists of three parts. The first part, referred to as SplitLine, involves dividing the line into two halves and determining whether each candidate occurs on the left or right side (some remain undetermined). The second part, referred to as SortCandidates, finds the order of candidates in each half and merges the sorted lists to obtain a total ordering. The final part, referred to as SortVoters, determines the ordering among the voters based on the retrieved order of candidates. Algorithm 1 demonstrates how these three components contribute to sorting the candidates and voters.

---

**ALGORITHM 1:** Sort candidates and voters

**Input:** Election instance $\mathcal{E}$.
**Output:** Sequence $S_C$ as the order of determined candidates and sequence $S_V$ as the order of voters.

1 **Function** SortCandidatesAndVoters($\mathcal{E} = (V, C, \succ)$):
2    $(L, R) \leftarrow$ SplitLine($\mathcal{E}$)
3    $S_C \leftarrow$ SortCandidates($\succ_1, L, R$)
4    $S_V \leftarrow$ SortVoters($\mathcal{E}, S_C$)
5    **return** $(S_C, S_V)$

---

**SplitLine.** In this part, we first find a pivot to split the line at that point. To achieve this, we introduce the following definitions. We arbitrarily choose the first voter as the pivot voter, associated with two pivot candidates as follows.

**Definition 24** (Pivot Voter and Candidates). The voter $v_1$ is called the *pivot voter*. Additionally, the two nearest candidates to the pivot voter are called the *pivot candidates*. Without loss of generality, assume that $c_1$ and $c_2$ are the two nearest candidates to the pivot voter $v_1$, with $c_1$ positioned to the left of $c_2$. Note that $c_1$ and $c_2$ may both be on the same side of $v_1$.

Finally, the point where the line is split is defined as follows:

**Definition 25** (Pivot Point). The midpoint of the line segment between the two pivot candidates is called the *pivot point*, denoted by $p$. Let $L$ and $R$ be the subsets of candidates on the left and right sides of $p$, respectively.

Finally, we aim to determine whether a candidate belongs to $L$ or $R$. Therefore, we formally define determined and undetermined candidates as follows:

**Definition 26** (Determined Candidates). A candidate is *determined* if it is known whether the candidate belongs to $L$ or $R$ based on Definition 25. Otherwise, the candidate is *undetermined*. Let $C^* = L \cup R$ be the set of determined candidates

Initially, based on Definition 24, we know that $c_1 \in L$ and $c_2 \in R$. Thus, $C^* = \{c_1, c_2\}$. Then, in multiple iterations, we expand $L$ and $R$ by adding as many candidates as possible. Therefore, at the end of each iteration, we update $C^*$ such that $C^* = L \cup R$ again. We also ensure that $C^*$ always forms a consecutive subset of candidates on the line. The process for determining new candidates in each iteration is as follows:

If there exists a voter $v_i$ such that two candidates $c_k$ and $c_j$ satisfy $c_k \succ_i c_j$, where $c_j \in C^*$ but $c_k \notin C^*$, then we can determine $c_k$'s membership as follows:

- If both $c_1$ and $c_2$ are closer to $v_i$ than $c_k$, then, if $c_j$ is in $L$, $c_k$ would be in $R$, and vice versa.

- Otherwise, $c_k$ would be in $L$ if $c_1$ is closer to $v_i$, and $c_k$ would be in $R$ if $c_2$ is closer to $v_i$.

Algorithm 2 is pseudocode for the function `Determine`, which determines a candidate if the above conditions hold and adds it to the corresponding set. Algorithm 3 determines as many candidates as possible in each iteration while maintaining the succession of the determined candidates (see A.1 for proofs). We call these candidates $C_{new}$. At the end of the iteration, it merges $C_{new}$ into $C^*$. It is important to note that we do not add each point immediately to $C^*$; instead, we merge them all at the end. This approach ensures that $C^*$ remains a consecutive list of candidates, which is crucial for our analysis.

The output of this procedure is $L$ and $R$, which represent the determined candidates on the left and right sides of the pivot point $p$, respectively.

---

**ALGORITHM 2:** Determine a candidate with a determined candidate in a voter's ordinal preference

---

**Input:** Ordinal preference of voter $v_i$, denoted by $\succ_i$; candidates $c_j$ and $c_k$ where $c_k \succ_i c_j$, $c_j$ is determined, but $c_k$ is not; two sets $L$ and $R$ containing currently determined candidates on the left and right sides of the pivot point $p$, respectively; and pivot candidates $c_1$ and $c_2$.

**Output:** Updated sets $L$ and $R$ including candidate $c_k$.

1 **Function** Determine($\succ_i, c_j, c_k, L, R, c_1, c_2$):
2     **if** $c_1 \succ_i c_k$ ***and*** $c_2 \succ_i c_k$ **then**
3         Add $c_k$ to $L$ if $c_j \in R$; otherwise, add $c_k$ to $R$
4     **else**
5         Add $c_k$ to $L$ if $c_1 \succ_i c_2$; otherwise, add $c_k$ to $R$
6     **end**
7     **return** $(L, R)$

---

**SortCandidates.** The goal of this part is to sort the candidates in $L$ and $R$. We know that $L$ lies to the left of $p$, and $R$ lies to the right of $p$, where $p$ is the midpoint of the segment connecting the

---

**ALGORITHM 3:** Determine Candidates

**Input:** Election instance $\mathcal{E}$.
**Output:** Two subsets $L$ and $R$ of candidates, where candidates are on the left and right of the pivot point $p$, respectively.

1 **Function** SplitLine($\mathcal{E} = (V, C, \succ)$):
2     $L \leftarrow \{c_1\}$
3     $R \leftarrow \{c_2\}$
4     $C^* \leftarrow \{c_1, c_2\}$
5     **repeat**
6        $C_{new} \leftarrow \emptyset$
7        **while** $\exists (v_i, c_j, c_k) : c_k \succ_i c_j$ **and** $c_k \notin C^*$ **and** $c_j \in C^*$ **do**
8           $(L, R) \leftarrow$ Determine($\succ_i, c_k, c_j, L, R, c_1, c_2$)
9           $C_{new} \leftarrow C_{new} \cup \{c_k\}$
10        **end**
11        $C^* \leftarrow C^* \cup C_{new}$
12     **until** $C_{new} = \emptyset$
13     **return** $(L, R)$

---

two nearest candidates of $v_1$ (see Definitions 24 and 25). Consequently, in the ordinal preference of $v_1$, candidates in $L$ with higher preferences are positioned to the right of those with lower preferences. Similarly, candidates in $R$ with higher preferences are positioned to the left of those with lower preferences. By combining these two observations, we can sort the candidates based on their positions along the line. Algorithm 4 presents the pseudocode for this approach.

---

**ALGORITHM 4:** Sort determined candidates

**Input:** Preference order of $v_1$, denoted as $\succ_1$, $L$ and $R$, candidates on the left and the right side of pivot $p$.
**Output:** Sequence $S_C$, sorted candidates in $L$ and $R$ by their position left to right.

1 **Function** SortCandidates($\succ_1, L, R$):
2     $S_C$ is an empty sequence of candidates
3     **for** $i$ *from* $1$ *to* $m$ **do**
4        Let $c_j$ be the $i$-th candidate in the preference order of $v_1$, i.e., $\succ_{1,i}$
5        **if** $c_j \in L$ **then**
6           Add $c_j$ to the extreme left of $S_C$
7        **else if** $c_j \in R$ **then**
8           Add $c_j$ to the extreme right of $S_C$
9        **end**
10     **end**
11     **return** $S_C$

---

**SortVoters.** This part focuses on sorting voters given the sorted determined candidates. The key observation is that a voter who prefers one candidate over another tends to be closer to the candidate they prefer more. Consequently, we have a method to compare two voters. For a pair of voters $v_i$ and $v_j$, assume $k$ is the smallest index where the ordinal preferences of $v_i$ and $v_j$ differ. $v_i$ is on the left side of $v_j$ if $\succ_{i,k}$ is on the left side of $\succ_{j,k}$. Recall that the notation $\succ_{p,q}$ represents the $q$-th candidate in the preference order of voter $v_p$. If $\succ_{i,k}$ and $\succ_{j,k}$ are not determined, assuming $v_i$ and $v_j$ are at the same position does not affect the further analysis (see Sections A.1). Algorithms 5 and 6 illustrates how function SortVoters works.

## A.1 ANALYSIS

This part focuses on verifying that the order of candidates and voters has been correctly calculated.

First, recall that the algorithm returns the sequence $S_C$ as the order of candidates (Line 3 of Algorithm 1). This sequence contains only determined candidates, denoted as $C^* = L \cup R$. Additionally, in Line 4 of Algorithm 1, we calculate the order of voters based on $S_C$.

It is important to note that for the remainder of this section, all lemmas consider a specific line metric election instance.

---

**ALGORITHM 5:** Compare two voters based on determined candidates

---

**Input:** Two voters $v_i$ and $v_j$, preference profile $\succ$, and order of determined candidates $S_C$.
**Output:** An integer in $[-1, 1]$: $-1$ if $v_i$ is left of $v_j$, $0$ if they share the same position, and $1$ if $v_i$ is right of $v_j$.

1 **Function** CompareVoters($v_i, v_j, \succ, S_C$):
2      Let $k$ be the min index where the preference orders of $v_i$ and $v_j$ differ, i.e., the smallest $k \in [m]$ such that $\succ_{i,k} \neq \succ_{j,k}$
3      **if** *no such $k$ exists **or** $\succ_{i,k}$ is not in $S_C$* **then**
4          **return** 0
5      **else if** *$\succ_{i,k}$ appears before $\succ_{j,k}$ in $S_C$* **then**
6          **return**-1
7      **else**
8          **return** 1
9      **end**

---

**ALGORITHM 6:** Sort voters

---

**Input:** Election instance $\mathcal{E}$, and order of determined candidates $S_C$.
**Output:** Sequence of all voters from left to right $S_V$.

1 **Function** SortVoters($\mathcal{E} = (V, C, \succ), S_C$):
2      Let $S_V$ be the sorted elements of $V$ using comparison function CompareVoters
3      **return** $S_V$

---

First, the following lemma demonstrates that a prefix of a voter's ordinal preference forms a consecutive subsequence of candidates.

**Lemma 27.** For any voter $v_i \in V$ and $1 \leq k \leq m$, the $k$ most preferred candidates of voter $v_i$ form a consecutive subsequence of candidates.

*Proof.* Assume the condition does not hold for a voter $v_i$ and some $k$. Since any single candidate forms a consecutive subsequence, we must have $k \geq 2$. Let $A$ denote the $k$ most preferred candidates by the voter $v_i$. By assumption, there exist two candidates $c_l, c_r \in A$ such that a candidate $c_m \in C \setminus A$ lies between them.

Assume without loss of generality that $c_l$ is to the left of $c_m$ and $c_r$ is to the right of $c_m$. Further, suppose $v_i$ is located to the left of $c_m$. Since $c_m \notin A$, we must have $c_r \succ_{v_i} c_m$. However, since the candidates are arranged in a line, it follows that $c_m \succ_{v_i} c_r$, which is a contradiction (illustrated in Figure 4). $\square$

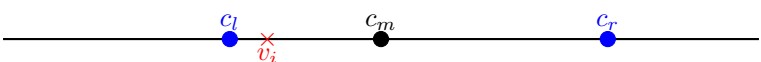

Figure 4: This figure is the illustration of the succession of nearest of any voter.

**Corollary 28.** $c_1$ and $c_2$ are consecutive.

Next, recalling the definitions of determined and undetermined candidates (Definition 26), the following lemma formally proves that the function Determine in Algorithm 2 correctly determines an undetermined candidate.

**Lemma 29.** Assume that $C^*$ is a consecutive subset of determined candidates, including $c_1$ and $c_2$. If there exists a voter $v_i$ and candidates $c_j \in C^*$ and $c_k \notin C^*$ such that $c_k \succ_i c_j$, then the function Determine correctly determines $c_k$.

*Proof.* Let us consider two cases regarding the positioning of $c_1$, $c_2$, and $c_k$ in the ordinal preference of $v_i$.

**Case 1:** Both $c_1$ and $c_2$ are positioned before $c_k$ in the ordinal preference of $v_i$.

Without loss of generality, we assume that $c_j$ is in $R$. We use proof by contradiction to show that $c_k$ is in $L$. Assume that $c_j$ is in $R$. Consider the prefix of the ordinal preference of $v_i$ ending with $c_k$. By Lemma 27, they must form a consecutive set of candidates. Therefore, $c_j$ must be on the right side of $c_k$. However, since $C^*$ consists of consecutive candidates, $c_k$ would necessarily be positioned on the right side of $c_j$. Because of the contradiction we conclude $c_k$ is in $L$ (illustrated in Figure 5)

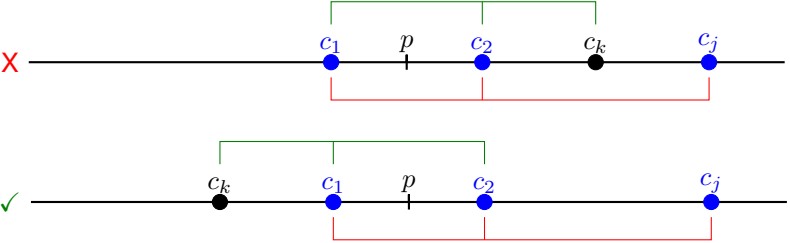

Figure 5: For a voter $v_i$, if we have $c_1 \succ_i c_k$, $c_2 \succ_i c_k$, and $c_k \succ_i c_k$, then $c_k$ and $c_j$ cannot both be in the same side of $p$. The figure above illustrates this contradiction, while the one below shows that they can be on opposite sides.

**Case 2:** At least one of $c_1$ and $c_2$ is positioned after $c_k$ in the ordinal preference of $v_i$.

Without loss of generality, assume that $c_1$ precedes $c_2$ in the ordinal preference of $v_i$. Consider the prefix of the ordinal preference of $v_i$ that contains $c_1$ and $c_k$ but not $c_2$. By Lemma 27, this prefix must form a consecutive sequence of candidates, meaning $c_2$ cannot lie between $c_1$ and $c_k$. If $c_k$ were in $R$, $c_2$ would lie between $c_1$ and $c_k$, as $c_1$ and $c_2$ are consecutive.

This contradiction implies that $c_k$ is in $L$ (illustrated in Figure 6).

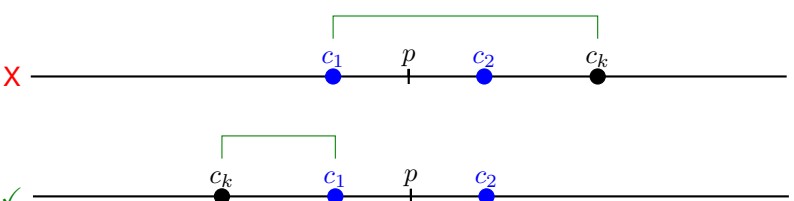

Figure 6: For a voter $v_i$, if we have $c_1 \succ_i c_2$ and $c_k \succ_i c_2$, then $c_k$ is in $L$. The figure above illustrates that if $c_k$ were in $R$, then $c_2$ would be in the consecutive subsequence of $c_1$ and $c_k$, which is a contradiction. On the other hand, the one below shows that $c_k$ must be in $L$.

$\square$

As explained in Lemma 29, determining a new candidate works if the current set of $C^*$ is consecutive. The following lemma shows that at the beginning of each iteration, $C^*$ is a consecutive subsequence of candidates.

**Lemma 30.** In Algorithm 3, at the beginning of each iteration (Line 5), all determined candidates, denoted as $C^*$, form a consecutive subsequence of all candidates.

*Proof.* Let $C_i^*$ be the set of determined candidates at the beginning of the $i$-th iteration. We prove that for any $i$, $C_i^*$ is a consecutive subsequence. $C_1^* = \{c_1, c_2\}$ satisfies the condition. (Corollary 28).

In the $(i-1)$-th iteration ($i \geq 2$), for any voter $v_j$, let $last_j$ be the least preferred candidate of $v_j$ in $C_{i-1}^*$. Then, by Lemma 29, by the end of iteration $i-1$, any candidate $c_k$ such that $c_k \succ_v last_j$ is determined. The set of candidates $c_k$ for which $c_k \succ_v last_j$ is denoted by $P_j$; note that $C_{i-1}^* \subseteq P_j$. As $P_j$ is a prefix of the ordinal preference of $v_j$, by Lemma 27, $P_j$ is a consecutive subsequence of candidates. Also $P_j$ contains $c_1$ and $c_2$ because $\{c_1, c_2\} \subseteq C_{i-1}^* \subseteq P_j$.

Since $C_i^* = C_{i-1}^* \cup C_{new}$ (Line 11) and $C_{new} = \bigcup_j (P_j \setminus C_{i-1}^*)$, we have $C_i^* = \bigcup_j P_j$. However, observe that a union of intersecting intervals is always an interval (this can be seen from a straightforward inductive argument). Since $P_j$ for all $j$ contain $\{c_1, c_2\}$, it follows that $C_i^*$ is a consecutive subsequence as well. □

Next, we show that $C^*$ is a core subset of candidates.

**Lemma 31.** By the end of the algorithm 3, $C^*$ is core, i.e., for any voter $v_i$, candidate $c_j \in C^*$, $c_k \notin C^*$, we have $c_j \succ_i c_k$.

*Proof.* Assume otherwise that $c_k \succ_i c_i$. Then, we can determine $c_k$ using $c_j$ and voter $v_i$ according to Lemma 29, and consequently, Algorithm 3 cannot have terminated at this point. Therefore, $c_j \succ_i c_i$ for any voter $v_i$. □

We now demonstrate that the function `SortCandidates` in Algorithm 4 returns the determined candidates in sorted order from left to right.

**Lemma 32.** The procedure `SortCandidates` correctly sorts determined candidates, denoted as $C^* = L \cup R$ from left to right.

*Proof.* Recall that sets $L$ and $R$ contain determined candidates (Definition 26). Now, consider the ordinal preference of $v_1$. For any pair of candidates $c_i, c_j \in L$, we show that $d(c_i, c_1) \leq d(c_j, c_1)$ if $c_i \succ_1 c_j$.

There are two cases:

- $c_1$ is on the right side of $v_1$ (Figure 7); we have:
$$d(c_i, c_1) - d(c_j, c_1) = d(c_i, v_1) - d(c_1, v_1) - d(c_j, v_1) + d(c_1, v_1)$$
$$= d(c_i, v_1) - d(c_j, v_1)$$

- $c_1$ is on the left side of $v_1$ (Figure 8); we have:
$$d(c_i, c_1) - d(c_j, c_1) = d(c_i, v_1) + d(c_1, v_1) - d(c_j, v_1) - d(c_1, v_1)$$
$$= d(c_i, v_1) - d(c_j, v_1)$$

Therefore, we have:
$$d(c_i, c_1) \leq d(c_j, c_1) \Leftrightarrow d(c_i, v_1) \leq d(c_j, v_1)$$

Based on the definition of ordinal preference, we have $d(c_i, v_1) \leq d(c_j, v_1)$ if $c_i \succ_1 c_j$. Consequently $d(c_i, c_1) \leq d(c_j, c_1)$ if $c_i \succ_1 c_j$.

Similarly, for any pair of candidates $c_i, c_j \in R$, we can show that $d(c_i, c_2) \leq d(c_j, c_2)$ if $c_i \succ_1 c_j$.

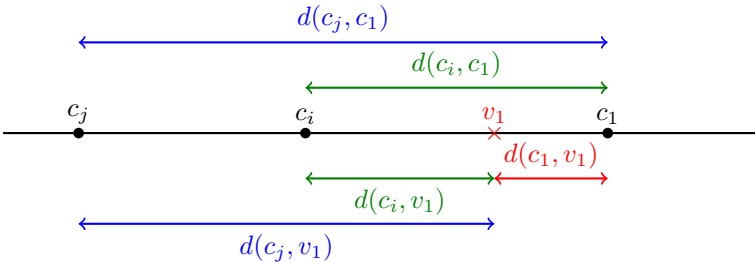

Figure 7: Illustration of the scenario where $c_1$ is positioned to the right of $v_1$.

The procedure `SortCandidates` processes candidates according to $\succ_1$, adding them to the left of $S_C$ if they are in $L$ and to the right otherwise. Therefore, $S_C$ contains candidates in the correct order from left to right. □

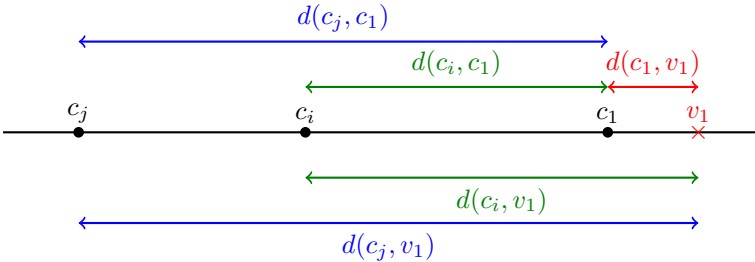

Figure 8: Illustration of the scenario where $c_1$ is positioned to the left of $v_1$.

Next, we show the correctness of the function `SortVoters`.

**Lemma 33.** The procedure `SortVoters` finds the order of voters with respect to the determined candidates.

*Proof.* By Lemma 32, we have the order among the determined candidates. Assume that for candidate $c_i$, $order_{c_i}$ is the index of $c_i$ from left to right. As `SplitLine` determines a prefix for each candidate, for a voter $v_i$, we let sequence $sorted_j = order_{\succ_{i,1}}, order_{\succ_{i,2}}, \ldots order_{\succ_{i,k}}$ where $k$ is the number of determined candidates. Now, we can compare voters' *sorted* sequences lexicographically. A voter with a smaller *sorted* is on the left side of a voter with a larger *sorted*. It is important to highlight that this approach may result in ties among some voters in the ordering. □

Next, we prove Theorem 23.

*Proof of Theorem 23.* By Lemma 31, Algorithm 1 first determines a core subset of candidates. Then, according to Lemma 32, it finds the correct order for this subset. Lastly, Lemma 33 demonstrates that the algorithm establishes a correct order with respect to the order of determined candidates. Therefore, as Algorithm 1 correctly identifies all three, the theorem follows. □

### A.2    FIND THE EXACT ORDER OF VOTERS

In this part, we extend the order result by proving that it is possible to determine the exact order of all voters, formally:

**Theorem 34.** For any election $\mathcal{E}$ consistent with a line metric, it is possible to determine the ordering of the voters along the line. Furthermore, one can derive an ordering of the candidates that is consistent with this metric.

Algorithm 1 determines the order for a core subset $C^*$ of candidates. Consequently, the first $|C^*|$ candidates in any voter's preference order are $C^*$. By examining the candidate at $|C^*|+1$-th position in each voter's ranking, we observe two cases.

- There exists a unique candidate $c_i$ such that every voter ranks $c_i$ in the $(|C^*|+1)$-th position.
- There exist two candidates, $c_i$ and $c_j$, such that for each voter, the candidate in the $(|C^*|+1)$-th position is either $c_i$ or $c_j$.

It is important to note that no more than two different candidates can appear in that position.

In the first case, $c_i$ does not create any distinctions between voters, as removing this column from the preference profile has no effect.

In the second case, we can make the following assumptions:

- First, if all voters, based on our prior knowledge, are in the same position, we assume $c_i$ is to the left of $c_j$. Otherwise, there are voters $v_p$ and $v_q$ for whom we know they are not in the same position, with one preferring $c_i$ and the other preferring $c_j$. This allows us to determine the order between $c_i$ and $c_j$.

- Second, when the candidates in $C^*$ are excluded, the candidates $c_i$ and $c_j$ appear consecutively on the line.

Excluding candidates in $C^*$, these two conditions enable us to re-run `SortCandidatesAndVoters`, leading to more determined candidates and a more accurate order of voters. Furthermore, this process can be extended by recursively running the algorithm on the remaining candidates until no candidates remain. At that point, any two voters who do not share the same preference orders will be distinguished, and the result will be the exact order of the voters.

Algorithm 7 outlines this approach in pseudo-code. In this pseudo-code, the notations $\succ^{(l)}$ and $\succ_i^{(l)}$ represent the preference order of all voters and voter $v_i$, respectively, starting from the $l$-th index. Additionally, $\circ$ denotes the concatenation of sequences, and set operators can be applied to sequences as well. It is also important to note that for `SplitLine`, we input $L$ and $R$ as the determined candidates at the beginning. This is not explained in Algorithm 3, but it can be applied by replacing the first three lines of Algorithm 3 with the input of $L$ and $R$ as the determined candidates.

---

**ALGORITHM 7:** Find the exact order of voters

**Input:** Election instance $\mathcal{E} = (V, C, \succ)$.
**Output:** Sequence $S_V$ as the exact order of voters.

1 **Function** `FindVotersOrder`$(\mathcal{E} = (V, C, \succ))$**:**
2   $(S_C, S_V) \leftarrow$ `SortCandidatesAndVoters`$(\succ_1)$
3   $C \leftarrow C - S_C$
4   $l \leftarrow |S_C| + 1$
5   **while** $C \neq \emptyset$ **do**
6     $C_{\text{first}} \leftarrow \{c_i \in C \mid \exists v_j \text{ such that } \succ_{j,l} = c_i\}$
7     **if** $|C_{\text{first}}| = 1$ **then**
8       $C \leftarrow C - C_{\text{first}}$
9       $l \leftarrow l + 1$
10       **continue**
11     **end**
12     $(c_i, c_j) \leftarrow C_{\text{first}}$
13     **if** *all voters in $S_V$ do **not** have equal positions* **then**
14       Let $v_p, v_q$ be two voters with different positions in $S_V$ such that $\succ_{p,l} = c_i$ and $\succ_{q,l} = c_j$
15       Swap $c_i$ and $c_j$ if $v_p$ is on the right side of $v_q$
16     **end**
17     $L \leftarrow \{c_i\}$
18     $R \leftarrow \{c_j\}$
19     $(L, R) \leftarrow$ `SplitLine`$((V, C, \succ^{(l)}), L, R)$
20     $S_{C'} \leftarrow$ `SortCandidates`$(\succ_1^{(l)}, L, R)$
21     $S_C \leftarrow (S_{C'} \cap L) \circ S_C \circ (S_{C'} \cap R)$
22     $S_V \leftarrow$ `SortVoters`$(\mathcal{E}, S_C)$
23     $C \leftarrow C - S_{C'}$
24     $l \leftarrow l + |S_{C'}|$
25   **end**
26   **return** $S_V$

---

**Analysis.** Now, we focus on the correctness of the approach explained earlier. Recalling the definition of core candidates (Definition 21), we define the covering range as follows:

**Definition 35** (Covering Interval). We define the range $[l, r]$ as the *covering interval* for an election $\mathcal{E} = (V, C, \succ)$ and a core subset of candidates $C^*$ if it is the smallest interval that includes $C^* \cup V$.

We first show that this interval does not contain any other candidates.

**Lemma 36.** For an election $\mathcal{E} = (V, C, \succ)$ and a core subset of candidates $C^*$, let $[l, r]$ be the covering interval. Any candidate $c_i \in C \setminus C^*$ lies outside the range $[l, r]$.

*Proof.* Assume, for the sake of contradiction, that there exists a candidate $c_i \notin C^*$ lying within the interval $[l, r]$. We consider three cases:

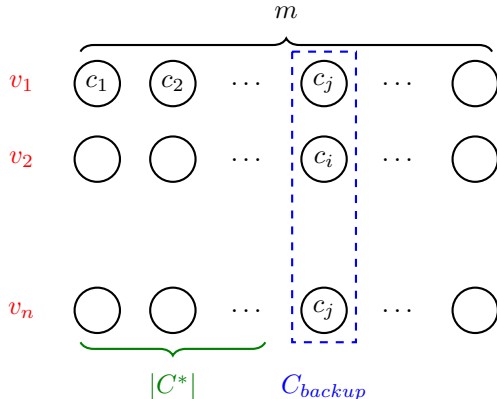

Figure 9: This figure illustrates the preference profile. Each row represents the preference order of a voter. The first $|C^*|$ cells in each row form the set $C^*$, while the candidates in the next column form the set $C_{backup}$.

- If all voters are on the left side of $c_i$, exist a $c_j \in C^*$ in range $(c_i, r]$. Then, any voter prefers $c_j$ to $c_i$, which contradicts $C^*$ to be a core.

- If all candidates in $C^*$ are on the right side of $c_i$, then there exists a voter $v_k$ in range $[l, c_i)$ such that $v_k$ prefers $c_i$ to all candidate in $C^*$ which is a contradiction.

- Otherwise. exists candidate $c_j \in C^*$ on the left side of $c_i$ and voter $v_j$ on the right side of $c_i$. we know $v_j$ prefers $c_i$ to $c_j$ which is a contradiction.

Consequently, all candidates in $C \setminus C^*$ are outside $[l, r]$. $\qquad \square$

Next, we assume the preference profile to be a table with $n$ rows and $m$ columns, where each row is the preference order of each voter, ordered by priority from left to right. Figure 9 illustrates.

Then, we define *backup* candidates as those appearing in the columns immediately following a core subset of candidates (Figure 9).

**Definition 37** (Backup Candidates). For an election $\mathcal{E} = (V, C, \succ)$ and a core subset of candidates $C^*$, let the *backup* subset of candidates be defined as

$$C_{backup} = \{c_i \in C \mid \exists v_j \text{ such that } \succ_{j, |C^*|+1} = c_i\}.$$

This means that a candidate $c_i$ belongs to $C_{backup}$ if there exists a voter $v_j$ who ranks $c_i$ in the $(|C^*| + 1)$-th position.

First, we show that a backup subset of candidates either has one or two candidates.

**Lemma 38.** For an election $\mathcal{E} = (V, C, \succ)$ and a core subset $C^*$ that does not cover $C$, let $C_{backup}$ be the backup subset of candidates. Then:

- $|C_{backup}| \in \{1, 2\}$, and

- if $|C_{backup}| = 2$, all candidates between the two elements of $C_{backup}$ form $C^*$.

*Proof.* Let $[l, r]$ be the interval including all $V \cup C^*$. By Lemma 36, no candidates of $C_{backup}$ is within interval $[l, r]$. Assume that on one side of the interval $[l, r]$, there exist two candidates of $C_{backup}$. We refer to them as $c_i$ and $c_j$ and assume, without loss of generality, that they are located in $(-\infty, l)$, and $c_j$ is to the left of $c_i$. Then, for any voter $v \in V$, $v$ would prefer $c_i$ to $c_j$ which contradicts $c_j$ being included in $C_{backup}$ as illustrated in Figure 10.

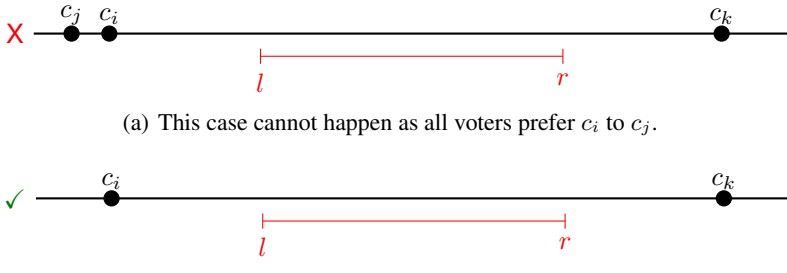

(a) This case cannot happen as all voters prefer $c_i$ to $c_j$.

(b) This case can happen without any issues.

Figure 10: Illustration on the number of elements in $C_{backup}$

Consequently, each side has at most one candidate in $C_{backup}$. Furthermore, by the same argument, no candidate in $C \setminus (C^* \cup C_{backup})$ can be positioned between the two candidates of $C_{backup}$. □

**Corollary 39.** For an election $\mathcal{E} = (V, C, \succ)$ and a core subset $C^*$ that does not cover $C$, let $C_{backup}$ be the backup subset of candidates that has two candidates. Then, $C_{backup}$ is a consecutive subsequence of $C \setminus C^*$

**Lemma 40.** For an election $\mathcal{E} = (V, C, \succ)$ and $C^*$ a core subset of candidates not covering $C$. Assuming $C_{backup}$ is the backup subset of candidates for $C^*$, If $|C_{backup}| = 2$ and the order of voters with respect to $C^*$ distinguishes at least two voters, we can determine the order of $C_{backup}$.

*Proof.* Based on Lemma 38, if $C_{backup} = \{c_i, c_j\}$, then each of them is on one end of the line. Let $V_l$ and $V_r$ be the set of leftmost and rightmost candidates with respect to $C^*$. As $C^*$ distinguishes at least two voters, $V_l \neq V_r$. The goal here is find $v_l \in V_l$ and $v_r \in V_r$ such that $v_l$ prefers one of $\{c_i, c_j\}$ while $v_r$ prefers the other. In this case, we can conclude which of $c_i$ and $c_r$ is on the left side of the other. Let

$$C^*_{backup} = \{c_i \in C \mid \exists v_j \in V_l \cup V_r \text{ such that } \succ_{j, |C^*|+1} = c_i\},$$

Now, it suffices to show that $|C^*_{backup}| = 2$, as we can easily choose $v_l$ and $v_r$. Suppose instead that $|C^*_{backup}| = 1$. Without loss of generality, let $C^*_{backup} = \{c_i\}$. Then, there exists a voter $v_m \notin V_l \cup V_r$ such that $\succ_{m, |C^*|+1} = c_j$. By Lemma 38, we know that $c_i$ and $c_j$ are different sides of $V_l$ and $V_r$. Additionally, $v_m$ is positioned between $V_l$ and $V_r$, which leads to a contradiction since both $V_l$ and $V_r$ prefer $c_i$. Figure 11 illustrates this.

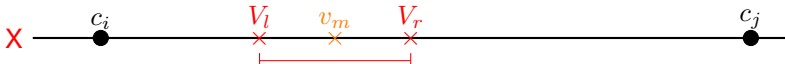

Figure 11: This figure illustrates that when $V_r$ prefers $c_i$ and $c_j$ is on the right side of $V_r$, voter $v_m$ cannot prefer $c_j$ over $c_i$.

□

Next, we focus on the order of voters in the case $|C^*| = 1$.

**Lemma 41.** For an election $\mathcal{E} = (V, C, \succ)$ and a core subset of candidates $C^*$, if $C_{backup}$ contains only one candidate $c_i$, then for any subset $A \subseteq C$ that does not include $c_i$, the ordering of voters with respect to $A$ is the same as the ordering of voters with respect to $A \cup \{c_i\}$.

*Proof.* Recalling the definition of "with respect to" (Definition 22), voters are compared based on the preference profile of the given subset in lexicographic order. Assume the order with respect to $A \cup \{c_i\}$. We know that $c_i$ has a fixed position in all voters' preference orders. Additionally, the set of candidates before this position is disjoint from the set of candidates after it.

Thus, comparing two voters involves two steps:

- First, we compare the portion of the preference order before $c_i$ lexicographically.

- If they are identical, we then compare the portion after $c_i$ lexicographically.

Due to this disjointness, the order with respect to $A$ follows the same structure. Consequently, the order of voters in both cases is identical. □

Finally, we provide the proof of Theorem 34

*Proof of Theorem 34.* We are showing that Algorithm 7 provides the exact order of voters. First, it runs `SortCandidatesAndVoters`, and according to Theorem 23, it determines the order of candidates for a core subset and retrieves the order of voters with respect to this order.

In each iteration of Line 5, by Lemma 38, the next position in the preference profile contains either one or two candidates.

If it contains one candidate:

- We can include this candidate in $C^*$, resulting in a larger core.

- Additionally, by Lemma 41, the position of this candidate does not affect the order of voters, so we can skip it in the order of this core subset.

On the other hand, if it contains two candidates:

- We determine their order using Lemma 40.

- Then, excluding $C^*$, Corollary 39 ensures that these two candidates are consecutive, allowing us to identify additional candidates using `SplitLine`.

- Next, we sort them based on the order of the first voter, as Lemma 36 and Lemma 38 indicate that the first voter is positioned between these two candidates.

- Furthermore, by concatenating the left and right parts of this newly ordered candidate set, we obtain a larger core with its sorted version.

- Lastly, we sort the voters based on the expanded ordering of candidates.

By the end of the algorithm, since the core is extended at each step, all effective candidates are considered in the voter ordering. Therefore, the exact order of voters is identified. □

## B  PROOFS OF UPPER BOUNDS ON DISTORTION FACTOR

### B.1  UPPER BOUNDS FOR SUM-COST

**Lemma 6.** When considering the sum of distances objective and the candidates and voters are located on the real line, one of the candidates directly to the right or left of the median voter will be an optimal candidate.

*Proof.* Denote the median voter as $v$. Assume the chosen candidate is not immediately to the left or right of $v$. Without loss of generality, let us consider that the chosen candidate $c$ lies to the left of $v$, and denote the first candidate to the left of $v$ as $c_l$. We now formally show that replacing $c$ with $c_l$ results in a solution that is at least as effective in minimizing voter distances.

The change in the total distance is given by:

$$\Delta = \sum_u (d(u, c_l) - d(u, c)) = \sum_{u \in L_v} (d(u, c_l) - d(u, c)) + \sum_{u \in R_v} (d(u, c_l) - d(u, c)),$$

where $L_v = \{u \in V \mid u < v\}$ represents voters to the left of $v$, and $R_v = \{u \in V \mid u \geq v\}$ represents voters to the right of $v$ (including $v$ itself).

For each voter $u \in R_v$, since both $c$ and $c_l$ are to the left of these voters, we have:

$$\forall u \in R_v, \quad d(u, c_l) = d(u, c) - d(c, c_l).$$

For each voter $u \in L_v$, we observe that:

$$\forall u \in L_v, \quad d(u, c_l) \leq d(u, c) + d(c, c_l).$$

Substituting these relations, the change in total distance can be bounded as:

$$\Delta \leq \sum_{u \in L_v} d(c, c_l) - \sum_{u \in R_v} d(c, c_l) = |L_v| \cdot d(c, c_l) - |R_v| \cdot d(c, c_l).$$

Since $v$ is the median voter and belongs to $R_v$, it follows that $|L_v| \leq |R_v|$. Thus:

$$\Delta \leq 0.$$

Therefore, replacing $c$ with $c_l$ does not increase the total distance, ensuring that choosing $c_l$ is at least as good as choosing $c$.

Since $v$ is the median voter, there are at least as many voters to the right of $v$ (including $v$) as there are to its left (excluding $v$). Therefore, the total cost cannot increase when moving from $c$ to $c_l$, implying that the cost of $c_l$ is *at most* the cost of $c$. Similarly, if a candidate $c_r$ directly to the right of $v$ exists, any candidate farther to the right will have a distance that is at least as great. Hence, either $c_l$ or $c_r$ will be an optimal candidate that minimizes the total cost. □

**Lemma 7.** There exists a voting rule for the 3-committee election on the line metric such that the resulting committee includes an optimal candidate.

*Proof.* First, we can determine the order of a subset of candidates $C^*$ and all voters based on the algorithms in Section A (see Theorem 23). Additionally, by Lemma 31, any candidate $c \notin C^*$ cannot be an optimal candidate, since $c$ will be farther from all voters than any candidate in $C^*$. Thus, we can ignore the candidates outside $C^*$ and proceed with the remaining candidates.

Now, we consider the median voter $v$ in the ordering of voters. We note that there might be a tie between multiple voters for this position, but we will only utilize the voter's closest candidate, which will be the same for all tied voters.

After identifying the median voter, we consider $v$'s closest candidate $c$. This candidate will be either the one immediately to the left or the right of the median voter $v$. Next, we use the ordering of the candidates to find the candidates $c_l$ to the left of $c$ and $c_r$ to the right of $c$. We claim that the set $\{c, c_l, c_r\}$ is guaranteed to include an optimal candidate: if $c$ is to the left of the median voter, then $c_r$ will be the first candidate to the right of the median, and one of $c$ or $c_r$ will be an optimal candidate by Lemma 6. Similarly, if $c$ is to the right of the median, one of $c$ or $c_l$ will be an optimal candidate. Therefore, the set $\{c, c_l, c_r\}$ will contain an optimal candidate (Figure 12).

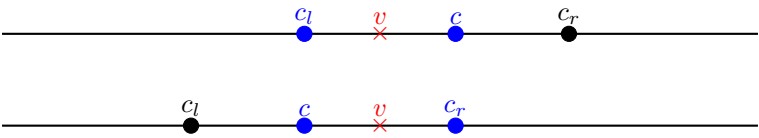

Figure 12: This figure illustrates that the set of three candidates—comprising the closest candidate to the median voter, as well as the candidates positioned to the left and right of this candidate—always includes the optimal candidate. Here, $v$ represents the median candidate, $c$ is the closest candidate to $v$, and $c_l$ and $c_r$ are the candidates on either side. The figure considers two cases based on the position of $c$ relative to $v$.

Finally, we note that candidates $c_l$ and $c_r$ might not exist. In such cases, the candidates to the left and right of $v$ are still selected if they exist. □

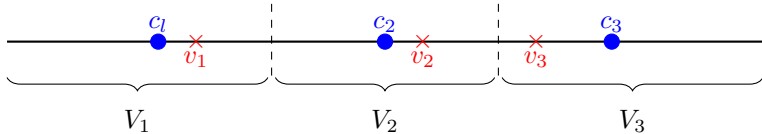

Figure 13: A figure illustrating three candidates $c_1$, $c_2$, and $c_3$ along with the possible locations of voters closest to each candidate, $V_1$, $V_2$, and $V_3$. Voters $v_1$, $v_2$, and $v_3$ show examples of voters in each set.

**Theorem 5.** There exists a voting rule for the 2-committee election for the sum-cost objective, on the line metric, such that the resulting committee includes an optimal candidate. Consequently, the 1-distortion of the voting rule is 1.

*Proof.* By Lemma 7, we can select three candidates that include an optimal candidate. Let $c_1$, $c_2$, and $c_3$ denote these candidates from left to right. We then define the sets $V_1$, $V_2$, and $V_3$ as the sets of voters who prefer the corresponding candidate to the other two, as illustrated in Figure 13. We note that the sets $V_1, V_2, V_3$ can be determined using ordinal preferences.

Thus, we can state that

$$
\begin{aligned}
\text{cost}_s(c_2) &= \sum_{v \in V} d(v, c_2) \\
&= \sum_{v \in V_1} d(v, c_2) + \sum_{v \in V_2} d(v, c_2) + \sum_{v \in V_3} d(v, c_2) \\
&\leq \sum_{v \in V_1} d(v, c_2) + \sum_{v \in V_2} d(v, c_2) + \sum_{v \in V_3} (d(v, c_3) + d(c_2, c_3)) \quad \text{(Triangle Inequality)} \\
&\leq \sum_{v \in V_1} d(v, c_2) + \sum_{v \in V_2} d(v, c_3) + \sum_{v \in V_3} (d(v, c_3) + d(c_2, c_3)) \\
&\qquad\qquad (\forall_{v \in V_2}\ d(v, c_2) \leq d(v, c_3)) \\
&= \sum_{v \in V_1} (d(v, c_3) - d(c_3, c_2)) + \sum_{v \in V_2} d(v, c_3) + \sum_{v \in V_3} (d(v, c_3) + d(c_2, c_3)) \\
&\qquad\qquad (\forall_{v \in V_1}\ d(v, c_2) = d(v, c_3) - d(c_2, c_3)) \\
&= \text{cost}_s(c_3) - |V_1| \cdot d(c_2, c_3) + |V_3| \cdot d(c_2, c_3) \\
&= \text{cost}_s(c_3) + (|V_3| - |V_1|) \cdot d(c_2, c_3).
\end{aligned}
$$

Similarly, we can show that $\text{cost}_s(c_2) \leq \text{cost}_s(c_1) + (|V_1| - |V_3|) \cdot d(c_2, c_1)$. Now, depending on whether $|V_1| < |V_3|$ or not, we can see that either $\text{cost}_s(c_2) \leq \text{cost}_s(c_1)$ or $\text{cost}_s(c_2) \leq \text{cost}_s(c_3)$. So we can disregard one of $c_1$ and $c_3$ based on this and still find the optimum, as $c_2$ is guaranteed to be a better candidate than the discarded one. $\qquad \square$

**Theorem 8.** There exists a voting rule choosing $m - 1$ out of $m$ candidates achieving a 1-distortion of $1 + \frac{2}{m-1}$ for the sum-cost objective.

*Proof.* For each voter $v$, let $\textit{first}_v$ be their closest candidate. Then, we claim that the voting rule that chooses the $m - 1$ candidates appearing most frequently in $\textit{first}$ achieves the desired distortion. For a given instance, let $C$ be the set of candidates selected by this algorithm and $c_{\text{opt}}$ be an (arbitrary) optimal single candidate. If $c_{\text{opt}} \in C$, then we get a 1-distortion of 1, and we are done. Otherwise, $C$ includes every candidate except for $c_{\text{opt}}$. Now, we can bound the optimal cost $\text{OPT}$ in this instance as

$$\text{OPT} = \sum_{i \in [n]} d(c_{\text{opt}}, v)$$

$$= \sum_{\substack{v \in V \\ \textit{first}_v = c_{\text{opt}}}} d(c_{\text{opt}}, v) + \sum_{\substack{v \in V \\ \textit{first}_v \neq c_{\text{opt}}}} d(c_{\text{opt}}, v)$$

$$\geq \sum_{\substack{v \in V \\ \textit{first}_v = c_{\text{opt}}}} d(c_{\text{opt}}, v) + \sum_{\substack{v \in V \\ \textit{first}_v \neq c_{\text{opt}}}} d(C, v). \tag{1}$$

where the last inequality follows since for any voter $v \in V$, $\textit{first}_v \in C$ if $\textit{first}_v \neq c_{\text{opt}}$. Let $v'$ be the voter closest to $c_{\text{opt}}$ such that $\textit{first}_v \neq c_{\text{opt}}$. Then, we can use this to state that

$$\text{OPT} \geq \sum_{\substack{v \in V \\ \textit{first}_v \neq c_{\text{opt}}}} d(c_{\text{opt}}, v)$$

$$\geq \sum_{\substack{v \in V \\ \textit{first}_v \neq c_{\text{opt}}}} d(c_{\text{opt}}, v')$$

$$\geq \left(n - \frac{n}{m}\right) d(c_{\text{opt}}, v') \qquad (\textit{first}_v = c_{\text{opt}} \text{ for at most } \frac{n}{m} \text{ voters based on choice of } C)$$

and therefore

$$\frac{n}{m} d(c_{\text{opt}}, v') \leq \frac{1}{m-1} \text{OPT}. \tag{2}$$

Now, we can bound the cost of $C$ as follows:

$$\text{cost}_m(C) = \sum_{v \in V} d(C, v)$$

$$= \sum_{\substack{v \in V \\ \textit{first}_v = c_{\text{opt}}}} d(C, v) + \sum_{\substack{v \in V \\ \textit{first}_v \neq c_{\text{opt}}}} d(C, v)$$

$$\leq \sum_{\substack{v \in V \\ \textit{first}_v = c_{\text{opt}}}} (d(c_{\text{opt}}, v) + d(c_{\text{opt}}, v') + d(C, v')) + \sum_{\substack{v \in V \\ \textit{first}_v \neq c_{\text{opt}}}} d(C, v)$$

$$\text{(Triangle inequality)}$$

$$\leq \sum_{\substack{v \in V \\ \textit{first}_v = c_{\text{opt}}}} (d(c_{\text{opt}}, v) + d(c_{\text{opt}}, v') + d(c_{\text{opt}}, v')) + \sum_{\substack{v \in V \\ \textit{first}_v \neq c_{\text{opt}}}} d(C, v)$$

$$(\textit{first}_{v'} \neq c_{\text{opt}} \text{ and } \textit{first}_{v'} \in C)$$

$$\leq \text{OPT} + \sum_{\substack{v \in V \\ \textit{first}_v = c_{\text{opt}}}} 2d(c_{\text{opt}}, v') \qquad \text{(By Equation 1)}$$

$$= \text{OPT} + 2 \cdot \frac{n}{m} d(c_{\text{opt}}, v') \qquad (\textit{first}_v = c_{\text{opt}} \text{ for at most } \frac{n}{m} \text{ voters})$$

$$\leq \left(1 + \frac{2}{m-1}\right) \text{OPT} \qquad \text{(By Equation 2)}$$

$\square$

## B.2 Upper bounds for max-cost

**Lemma 9.** *If the distance between $v_l$ and $v_r$ is $D$ (i.e., $D = d(v_l, v_r)$), then the cost for the optimal single candidate* OPT *satisfies* $\text{OPT} \geq \frac{D}{2}$.

*Proof.* Let $c_{\text{opt}}$ be an optimal candidate. The distance between $v_l$ and $c_{\text{opt}}$ is $d(v_l, c_{\text{opt}})$, and the distance between $v_r$ and $c_{\text{opt}}$ is $d(v_r, c_{\text{opt}})$. By the triangle inequality:

$$d(v_l, c_{\text{opt}}) + d(v_r, c_{\text{opt}}) \geq D.$$

Thus,

$$\max(d(v_l, c_{\text{opt}}), d(v_r, c_{\text{opt}})) \geq \frac{D}{2}.$$

Therefore, $\texttt{OPT} \geq \frac{D}{2}$. $\qquad\square$

**Lemma 10.** If there are no candidates placed between $v_l$ and $v_r$, a voting rule that selects $c_l$ and $c_r$, achieves a $1 - distortion$ of 1.

*Proof.* In this case, the voters are next to each other in a block, with $c_l$ being the first candidate immediately to the left of all voters and $c_r$ being the first candidate immediately to the right of all voters. So, by selecting $c_l$ and $c_r$, we ensure that the closest candidate to each voter is included. Therefore, the cost of this set is, at most, that of the single optimal candidate, and thus we conclude that $1 - distortion$ is 1. $\qquad\square$

**Lemma 11.** If $d(v_l, v_r) = D$, then for every voter $v$, either $d(v, v_l) \leq \texttt{OPT}$ or $d(v, v_r) \leq \texttt{OPT}$.

*Proof.* Since $v$ is a voter between $v_l$ and $v_r$ (recall $v_l$ is the left-most and $v_r$ is the right-most voter), we have:

$$d(v_l, v) + d(v, v_r) = D.$$

Thus, either $d(v_l, v) \leq \frac{D}{2}$ or $d(v, v_r) \leq \frac{D}{2}$.

By Lemma 9, $\frac{D}{2} \leq \texttt{OPT}$. Therefore, we conclude:

$$d(v_l, v) \leq \frac{D}{2} \leq \texttt{OPT}, \quad \text{or} \quad d(v, v_r) \leq \frac{D}{2} \leq \texttt{OPT}.$$

$\qquad\square$

**Theorem 12.** For any election $\mathcal{E}$ on the line metric, let $v_l$ and $v_r$ be the leftmost and rightmost voters, and $c_l$ and $c_r$ be the closest candidates to $v_l$ and $v_r$, respectively. Then, a voting rule $f$ that outputs the set $\{c_l, c_r\}$ satisfies $\texttt{1-distortion}(f) \leq 2$ with respect to the max-cost objective.

*Proof.* Since $c_l$ is closest to $v_l$ and we know $d(v_l, c_{\text{opt}}) \leq \texttt{OPT}$, it follows that:

$$d(v_l, c_l) \leq d(v_l, c_{\text{opt}}) \leq \texttt{OPT}.$$

Similarly, $d(v_r, c_r) \leq \texttt{OPT}$.

For any voter $v$, by Lemma 11, either $d(v, v_l) \leq \texttt{OPT}$ or $d(v, v_r) \leq \texttt{OPT}$. Without loss of generality, assume that $d(v, v_l) \leq \texttt{OPT}$. Then:

$$d(v, c_l) \leq d(v, v_l) + d(v_l, c_l) \leq \texttt{OPT} + \texttt{OPT} = 2 \cdot \texttt{OPT}.$$

Similarly, in the other case:

$$d(v, c_r) \leq 2 \cdot \texttt{OPT}.$$

Thus, for every voter, the distance to the closest candidate in $\{c_l, c_r\}$ is at most $2 \cdot \texttt{OPT}$. Hence, this selection achieves a distortion of 2. $\qquad\square$

**Theorem 13.** For any election $\mathcal{E}$ on the line metric, there exists a voting rule to select three candidates, which achieves a 1-distortion of $3/2$ with respect to the max-cost objective.

*Proof.* Let $v_l$ and $v_r$ be the leftmost and rightmost voters, and $c_l$ and $c_r$ their closest candidates. Using the ordering of the candidates, let $c_r'$ be the candidate immediately to the left of $c_r$. We claim that the set $C = \{c_l, c_r, c_r'\}$ achieves a distortion of $3/2$.

First, we note that any voter to the left of $c_l$ or the right of $c_r'$ has their closest candidate included in this set. Specifically, for any voter that is either left of $c_l$ or right of $c_r$, the closest candidates are $c_l$ and $c_r$, respectively. For candidates between $c_r'$ and $c_r$, the closest candidate is either $c_r$ or $c_r'$.

Now, one of $c_r'$ or $c_r$ must be the rightmost candidate between $v_l$ and $v_r$. Let this be candidate $c_r^* \in \{c_r, c_r'\}$. We already know that any voter outside the interval between $c_l$ and $c_r^*$ has their closest candidate included. In addition, we have

$$d(c_l, c_r^*) \leq d(c_l, v_l) + d(v_l, c_{\text{opt}}) + d(c_{\text{opt}}, c_r^*) \leq 3\texttt{OPT}.$$

Therefore, for any voter $v$ in this interval,

$$\min\{d(v, c_l), d(v, c_r^*)\} \leq \frac{3}{2}\text{OPT}.$$

So, the maximum distance of any voter to the candidate set $C$ is at most $3\text{OPT}/2$, and we achieve a $1 - distortion$ of $3/2$. $\square$

**Theorem 14.** For any election $\mathcal{E}$ on the line metric, there exists a voting rule to select four candidates, which achieves a 1-distortion of 1 with respect to the max-cost objective.

*Proof.* Again, we consider the leftmost and rightmost voters $v_l$ and $v_r$, and their closest candidates $c_l$ and $c_r$. In addition, we choose candidate $c_l'$ to the right of $c_l$ and $c_r'$ to the left of $c_r$ based on the ordering of candidates. Now, one of $c_l$ and $c_l'$ will be the first candidate to the right of $v_l$: If $c_l$ is to the right of $v_l$, it must be the first such candidate. Otherwise, it is the first candidate to the left of $v_l$, so $c_l'$ is to the right of $v_l$. Let this candidate be $c_l^*$ and define $c_r^*$ similarly. Now, any voter $v$ to the left of $c_l'$ has their closest candidate included in the set, as $v$ is either between $c_l$ and $c_l'$, so one of these two is $v$'s closest candidate or $v$ is to the left of $c_l$, in which case $c_l$ must be $v$'s closest candidate. Similarly, the closest candidate to each voter to the right of $c_r'$ is selected (Figure 14.

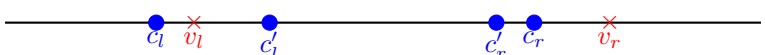

Figure 14: In this figure, the set $c_l, c_l', c_r, c_r'$ has a 1-distortion of 1. Here, $v_l$ and $v_r$ represent the leftmost and rightmost voters, respectively. For $v_l$, the closest candidate is $c_l$, which is positioned to its left; therefore, $c_l'$ is the closest candidate to its right. Conversely, for $v_r$, the closest candidate is $c_r$, which is also the closest candidate to its left, while $c_r'$ is the next leftward candidate. However, in this case, $c_r'$ is not necessary to achieve a 1-distortion of 1.

Next, since both $c_l^*$ and $c_r^*$ are between $v_l$ and $v_r$ and $d(v_l, v_r) \leq 2\text{OPT}$, we have $d(c_l^*, c_r^*) \leq 2\text{OPT}$. Therefore, any voter between $c_l^*$ and $c_r^*$ has a distance of at most $\text{OPT}$ to the closer candidate in $\{c_l^*, c_r^*\}$. As every voter is either between $c_l^*$ and $c_r^*$ or outside the interval $c_l', c_r'$, the distance of each voter to the closest candidate in our selected set is at most $\text{OPT}$, and we get a distortion of 1. $\square$

## C PROOFS OF LOWER BOUND ON DISTORTION FACTOR

### C.1 LOWER BOUND FOR SUM-COST

**Theorem 15.** For $k \geq 3$, there exists an instance for the $k$-committee election (with respect to the sum-cost) where no voting rule choosing $k$ candidates can achieve a bounded distortion even when compared to an optimal choice of $\lceil \log_2(k+1) \rceil + 1$ candidates.

*Proof.* Without loss of generality, we assume that $k$ is of the form $2^\ell - 1$, and $\lceil \log_2(k+1) \rceil + 1 = \ell + 1$. Now, we consider an instance with $2^\ell$ candidates and voters on a line. We further assume that voters and candidates are numbered from 0 to $2^\ell - 1$ and use $b_i$ to denote the $\ell$ digit binary representation of $i$ for $0 \leq i < 2^\ell$. For each $b_i$, we assume the bits are numbered left to right from 0 to $\ell - 1$ with the $j$-th bit shown by $b_{i,j}$.

For any two values $0 \leq i, j < 2^\ell$, let $\text{LCP}_{i,j}$ be the length of the longest common prefix between $b_i$ and $b_j$. Now, for each voter $v_i$, the preference of $v_i$ is determined as follows: For any two candidates $c_r$ and $c_s$, if $\text{LCP}_{i,r} \neq \text{LCP}_{i,s}$, $v_i$ prefers the candidate with the longer common prefix. Otherwise, let $bit_1$ be the first $bit$ in $b_i$ that differs from $b_r$ and $b_s$. Similarly, let $bit_2$ be the first bit of $b_r$ that differs from $b_s$. Then, if $bit_1 = bit_2$, $v_i$ will prefer candidate $c_r$ to candidate $c_s$; otherwise, $c_s$ to $c_r$.

Now, any voting rule choosing $k = 2^\ell - 1$ candidates will not choose one of the candidates. Let $c_{i^*}$ be the candidate left unselected. We show that we can arrange the voters and candidates on the line such that this selection will have unbounded distortion compared to a committee containing only $\ell + 1$ candidates.

In our construction, we will place each candidate $c_i$ and voter $v_i$ on the same point and describe the position for each candidate. For each candidate $c_i$, let $len_i = \text{LCP}_{i,i*}$ be the length of the longest common prefix between $b_i$ and $b_{i*}$. For any candidate $c_i$ and index $0 \leq j \leq \ell - 1$, let $h_{i,j}$ equal $b_{i,j}3^{-j}$ if $j \leq len_i$; and $\varepsilon b_{i,j}3^{-j}$ otherwise, where $0 < \varepsilon < 1$ is a small arbitrary constant (to be set appropriately). Then, we will place candidate $c_i$ at

$$p_i = \sum_{j=0}^{\ell-1} h_{i,j} = \sum_{j=0}^{len_i} b_{i,j}3^{-j} + \sum_{j=len_i+1}^{\ell-1} \varepsilon b_{i,j}3^{-j}.$$

Now, we show that this placement follows the voters' preference orders. First, we note that candidates (and voters) are positioned on the line in increasing order of their indices. To show this, consider candidates $c_i$ and $c_j$ with $i < j$. Consider the index $ind = \text{LCP}_{i,j}$ where $b_i$ and $b_j$ first differ. Since $i < j$, we must have $b_{i,ind} = 0$ and $b_{j,ind} = 1$. Now, $h_{i,t} = h_{j,t}$ for $t < ind$, so we have

$$p_j \geq \sum_{t=0}^{ind-1} h_{j,t} + h_{j,ind}$$

while

$$p_i \leq \sum_{t=0}^{ind-1} h_{j,t} + \sum_{t=ind+1}^{\ell-1} h_{i,t}.$$

So, it suffices to show that $h_{j,ind} \geq \sum_{t=ind+1}^{\ell-1} h_{i,t}$. If $ind \leq len_j$, we would have $h_{j,ind} = 3^{-ind}$ while

$$\sum_{t=ind+1}^{\ell-1} h_{i,t} \leq \sum_{t=ind+1}^{\ell-1} 3^{-t} \leq \frac{3^{-ind}}{2}$$

so the inequality holds. Otherwise, if $len_j \leq ind$, we will also have $len_i \leq ind$, as $b_i$ and $b_j$ match in the first $ind$ indices. Therefore, we will have $h_{j,ind} \geq \varepsilon 3^{-ind}$ while

$$\sum_{t=ind+1}^{\ell-1} h_{i,t} \leq \sum_{t=ind+1}^{\ell-1} 3^{-t} \leq \varepsilon \frac{3^{-ind}}{2}.$$

In either case, we have shown that $p_i \leq p_j$.

Now, take voter $v_i$ and candidates $c_r$ and $c_s$ such that $v_i$ prefers $c_r$ to $c_s$. First, we consider the case where $\text{LCP}_{i,r} \neq \text{LCP}_{i,s}$. Since $v_i$ prefers $c_r$ to $c_s$, we must have $\text{LCP}_{i,r} > \text{LCP}_{i,s}$ if they are not equal. Then, we have

$$d(v_i, c_r) = |p_i - p_r| = \Big| \sum_{t=0}^{\ell-1} (h_{i,t} - h_{r,t}) \Big| \leq \sum_{t=\text{LCP}_{i,r}}^{\ell-1} |h_{i,t} - h_{r,t}|$$

while

$$d(v_i, c_s) = |p_i - p_s| = \Big| \sum_{t=0}^{\ell-1} (h_{i,t} - h_{s,t}) \Big| = \Big| \sum_{t=\text{LCP}_{i,s}}^{\ell-1} (h_{i,t} - h_{s,t}) \Big| \geq \big| h_{i,\text{LCP}_{i,s}} - h_{s,\text{LCP}_{i,s}} \big| - \sum_{t=\text{LCP}_{i,s}+1}^{\ell-1} h_{s,t}.$$

If $len_i \geq \text{LCP}_{i,s}$, then we will also have $len_s \geq \text{LCP}_{i,s}$, and since $b_i$ and $b_s$ differ at index $\text{LCP}_{i,s}$, one of $h_{i,\text{LCP}_{i,s}}$ and $h_{s,\text{LCP}_{i,s}}$ is 0 while the other is $3^{-\text{LCP}_{i,s}}$, so

$$d(v_i, c_s) \geq \big| h_{i,\text{LCP}_{i,s}} - h_{s,\text{LCP}_{i,s}} \big| - \sum_{t=\text{LCP}_{i,s}+1}^{\ell-1} h_{s,t} > 3^{-\text{LCP}_{i,s}} - 3^{-\text{LCP}_{i,s}}/2 = 3^{-\text{LCP}_{i,s}}/2.$$

Additionally, we have

$$d(v_i, c_r) \leq \sum_{t=\text{LCP}_{i,r}}^{\ell-1} |h_{i,t} - h_{r,t}| < 3^{-\text{LCP}_{i,r}+1}/2 \leq 3^{-\text{LCP}_{i,s}}/2 < d(v_i, c_s)$$

as $\text{LCP}_{i,r} > \text{LCP}_{i,s}$.

On the other hand, if $len_i \leq \text{LCP}_{i,s}$, we will have $len_s \leq \text{LCP}_{i,s}$ and $len_r \leq \text{LCP}_{i,s}$. Therefore, we get

$$d(v_i, c_s) \geq \left| h_{i,\text{LCP}_{i,s}} - h_{s,\text{LCP}_{i,s}} \right| - \sum_{t=\text{LCP}_{i,s}+1}^{\ell-1} h_{s,t} > \varepsilon 3^{-\text{LCP}_{i,s}} - \varepsilon 3^{-\text{LCP}_{i,s}}/2 = \varepsilon 3^{-\text{LCP}_{i,s}}/2$$

and

$$d(v_i, c_r) \leq \sum_{t=\text{LCP}_{i,r}}^{\ell-1} \left| h_{i,t} - h_{r,t} \right| < \varepsilon 3^{-\text{LCP}(i,r)+1}/2 \leq \varepsilon 3^{-\text{LCP}_{i,s}}/2 < d(v_i, c_s).$$

Finally, if $\text{LCP}_{i,r} = \text{LCP}_{i,s}$, both $c_r$ and $c_s$ will fall on the same side of $v_i$. If $b_{i,\text{LCP}_{i,r}} = 0$, then both $c_r$ and $c_s$ will be to the right of $v_i$, and since $c_r \succ_i c_s$, we must have $b_{r,\text{LCP}_{r,s}} = 0$ while $b_{s,\text{LCP}r,s} = 1$. Therefore, $r < s$ and $c_r$ will be to the left of $c_s$ and closer to $v_i$. Similarly, if $b_{i,\text{LCP}_{i,r}} = 1$, we will have $r > s$ and $c_r$ will again be closer to $v_i$ than $c_s$. So, the preferences of each voter $v_i$ are respected by this positioning.

Next, we show that the ratio of the cost of the selection excluding $c_{i*}$ to the cost of the optimal committee with $\ell + 1$ candidates is unbounded. Consider any candidate $c_j$ other than $c_{i*}$. Then, we have

$$d(v_{i*}, c_j) = \left| \sum_{t=0}^{\ell-1}(h_{i*,t} - h_{j,t}) \right| = \left| \sum_{t=len_j}^{\ell-1}(h_{i*,t} - h_{j,t}) \right|$$

$$\geq \left| h_{i*,len_j} - h_{j,len_j} \right| - \sum_{t=len_j+1}^{\ell-1} h_{j,t} \geq 3^{-len_j} - \varepsilon 3^{-len_j}/2 \geq 3^{-\ell}/2.$$

So the cost of the selection excluding $c_{i*}$ is at least $3^{-\ell}/2$, a constant independent of $\varepsilon$. On the other hand, we consider the following selection of at most $\ell + 1$ candidates. For each $0 \leq j < \ell$, consider the candidate $c_{i_j}$ such that $b_{i_j,t} = b_{i*,t}$ for $t < j$, $b_{i_j,j} = 1 - b_{i*,j}$ and $b_{i_j,t} = 0$ for all $t > j$. We show that the cost of the selection $\{c_{i_j} \mid 0 \leq j < \ell\} \cup \{c_{i*}\}$ is a multiple of $\varepsilon$.

First, it is clear that $v_{i*}$ has cost 0 given this selection, as $c_{i*}$ is included. Next, take any other voter $v_i$ and consider the candidate $c_{i_j}$ in our selection for $j = len_i$. Then, we have

$$d(v_i, c_{i_j}) = \left| \sum_{t=0}^{\ell-1}(h_{i,t} - h_{i_j,t}) \right| \leq \sum_{t=len_i+1}^{\ell-1} \left| h_{i,t} - h_{i_j,t} \right| \leq \sum_{t=len_i+1}^{\ell-1} \varepsilon 3^{-t} \leq \varepsilon 3^{-len_i}/2.$$

Now, since $\varepsilon > 0$ is a value we choose, we can arbitrarily decrease the cost of this selection while the cost of the selection excluding $c_{i*}$ remains a constant value of at least $3^{-\ell}/2$. Therefore, the distortion of any voting rule choosing $2^\ell - 1$ candidates is unbounded compared to the optimal selection of $\ell + 1$ candidates. $\square$

**Theorem 16.** For any number of candidates $m$ and any $\varepsilon > 0$, there exist instances of the $(m-1)$-committee election problem in the 2D Euclidean metric for which no voting rule can guarantee a 1-distortion factor less than $1 + \frac{2}{m-1} - \varepsilon$.

*Proof.* We consider a family of $m + 1$ instances $I_0, I_1, \cdots, I_m$ with $m$ voters $\{v_1, v_2, \cdots, v_m\}$ and $m$ candidates $\{c_1, c_2, \cdots, c_m\}$ on a two-dimensional plane. In terms of voters' and candidates' locations on the plane, each of these instances is distinct. However, in terms of voters' preference orders on candidates (i.e., ordinal information), $I_0, I_1, \cdots, I_m$ are the same. To be more specific, let us describe these instances.

Let $\ell = 3/\varepsilon - 1$. We first describe the instance $I_0$. For each $i \in [m]$, the candidate $c_i$ is located at the point $\left(-\ell, 2^{i-1}/2^m\right)$. For each $i \in [m]$, the voter $v_i$ is located at the point $\left(0, 2^{i-1}/2^m\right)$. (See Figure 15.) It is straightforward to observe that, for each $i \in [m]$, the preference order of the voter $v_i$ is

$$c_i \succ_i c_{i-1} \succ_i \cdots \succ_i c_1 \succ_i c_{i+1} \succ_i c_{i+2} \succ_i \cdots \succ_i c_m.$$

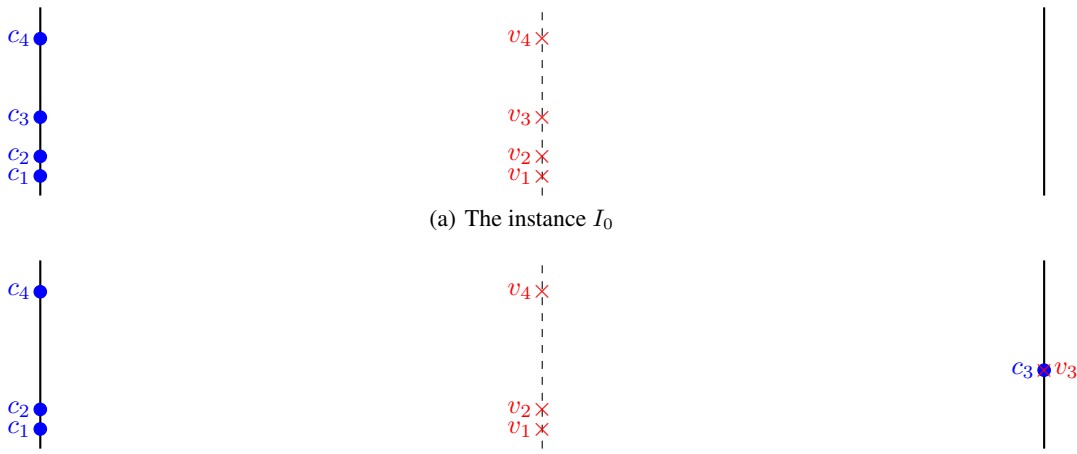

(a) The instance $I_0$

(b) $I_3$: Instance with voter $v_3$ and candidate $c_3$ moved

Figure 15: Figures of the lower bound instances. In (a), all candidates are located on the line $x = -\ell$, with voters with matching $y$-coordinates on the line $x = 0$. In (b), voter $v_3$ and candidate $c_3$ are moved to the line $x = \ell$, while keeping the same $y$-coordinate.

Next, for each $j \in [m]$, we define the instance $I_j$ as follows: For each $i \neq j \in [m]$, the locations of the candidate $c_i$ and the voter $v_i$ are the same as that in the instance $I_0$. Both the candidate $c_j$ and the voter $v_j$ are located at the point $\left(\ell, 2^{i-1}/2^m\right)$. (See Figure 15.) It is easy to observe that for any voter $v_i$ (where $i \neq j$), the distance to any candidate $c_j$ remains the same (as in $I_0$). Since the locations of other candidates remain the same, the preference order for any voter $v_i$ (where $i \neq j$) remains the same as in $I_0$. Let us now consider the voter $v_j$. Note that its closest candidate is $c_j$, and the distance to any other candidate $c_i$ remains the same as in $I_0$. Thus, the preference order of the voter $v_j$ is

$$c_j \succ_j c_{j-1} \succ_j \cdots \succ_j c_1 \succ_j c_{j+1} \succ_j c_{j+2} \succ_j \cdots \succ_j c_m$$

which is the same as that in $I_0$.

Let us now consider any arbitrary deterministic algorithm ALG that selects at most $m-1$ candidates. Now, suppose given the preference orders of voters as in $I_0$ (the same as in $I_1, \cdots, I_m$), ALG selects a set $C$ of candidates where $|C| \leq m - 1$. Suppose $c_k \notin C$, for some $k \in [m]$. Now, consider the instance $I_k$. Note, since voters' preference orders, i.e., ordinal information, are the same as in $I_0$, ALG also selects the set $C$ for the instance $I_k$. Then

$$\mathrm{cost}_s(C, I_k) = \sum_{i \in [m]} d(v_i, C)$$
$$\geq \sum_{i \neq k} \ell + 2\ell = (m+1)\ell.$$

On the other hand, selecting only the candidate $c_k$ for the instance $I_k$ would lead to the cost of

$$\mathrm{cost}_s(c_k, I_k) = \sum_{i \in [m]} ||v_i - c_k||_2 \leq \sum_{i \neq k}(\ell + 1) = (m-1)(\ell + 1)$$

and thus the optimal cost $\mathrm{OPT}(I_k) \leq (m-1)(\ell + 1)$.

Hence, the distortion factor of `ALG` on the instance $I_k$ is

$$
\begin{aligned}
\texttt{1-distortion(ALG)} &= \frac{\text{cost}_s(C, I_k)}{\text{OPT}(I_k)} \\
&\geq \frac{(m+1)\ell}{(m-1)(\ell+1)} \\
&= \left(1 + \frac{2}{m-1}\right)\left(1 - \frac{1}{\ell+1}\right) \\
&= 1 + \frac{2}{m-1} - \frac{1}{\ell+1}\left(1 + \frac{2}{m-1}\right) \\
&\geq 1 + \frac{2}{m-1} - \varepsilon \qquad\qquad \text{(since } \ell = 3/\varepsilon - 1\text{).}
\end{aligned}
$$

$\square$

**Theorem 17.** For any number of candidates $m$ and any $\varepsilon > 0$, there exist instances of the $(m-1)$-committee election problem with tree metrics for which no voting rule can guarantee a 1-distortion factor less than $1 + \frac{2}{m-1} - \varepsilon$.

*Proof.* Consider a star with a central vertex and $m$ leaves, such that the edge to leaf $i$ has cost $1 - \varepsilon_i$, where $\varepsilon \leq \varepsilon_i \leq 2\varepsilon$ are $m$ distinct values. Then, we construct $m$ instances on this tree, such that any voting rule has a 1-distortion of $1 + \frac{2}{m-1} - \varepsilon'$ on at least one instance. In every instance, there are $m$ candidates, with candidate $c_i$ located at leaf $i$. In addition, there are $m$ voters with voter $v_i$ located on the edge going toward leaf $i$. In the $i$-th instance, every voter except voter $v_i$ is located at distance $\varepsilon_i$ from the center of the star, while voter $v_i$ coincides with candidate $c_i$ on the leaf. This is illustrated in Figure 16.

In every instance, for any voter $v_j$, the distance to $c_j$ is at most $1 - \varepsilon_j - \varepsilon_i \leq 1 - 2\varepsilon$, while the distance to any other candidate $c_k$ is at least $1 - \varepsilon_k + \varepsilon_i \geq 1 - \varepsilon$. Therefore, the closest candidate to each voter $v_j$ is candidate $c_j$. For other candidates, the distances are determined by $1 - \varepsilon_k$, and each voter $v_j$'s preference order for candidates except its most preferred choice of $c_j$ will be in descending order of $\varepsilon_k$.

Therefore, the voters' preference orders are identical across the $m$ instances, and it is not possible to distinguish between the instances given the voters' ordinal preferences. Now, any deterministic voting rule choosing $m-1$ candidates must exclude one of the candidates. Let this candidate be $c_{i^*}$. Then, in instance $i^*$, the cost achieved by this voting rule is at least

$$
\sum_{j \neq i^*}(1 - \varepsilon_j - \varepsilon_i) + 1 - \varepsilon_{i^*} + 1 - \max_{j \neq i^*}\varepsilon_j \geq m - 1 + 2 - (2m)(2\varepsilon) = m + 1 - 4m\varepsilon
$$

while the cost of choosing only candidate $i^*$ is

$$
\sum_{j \neq i^*}(1 - \varepsilon_{i^*} + \varepsilon_{i^*}) = m - 1.
$$

Therefore, the 1-distortion of any voting rule can be lower bounded by

$$
\frac{m + 1 - 4m\varepsilon}{(m-1)} \geq 1 + \frac{2}{m-1} - 8\varepsilon.
$$

This completes the proof with $\varepsilon' = 8\varepsilon$. $\square$

## C.2 LOWER BOUNDS FOR THE MAX-COST

**Theorem 18.** Any deterministic algorithm for the $k$-committee election (with respect to the max-cost) that selects at most $k < m$ candidates out of $m$ candidates must have a 1-distortion of at least $3 - \varepsilon$ for any $\varepsilon > 0$.

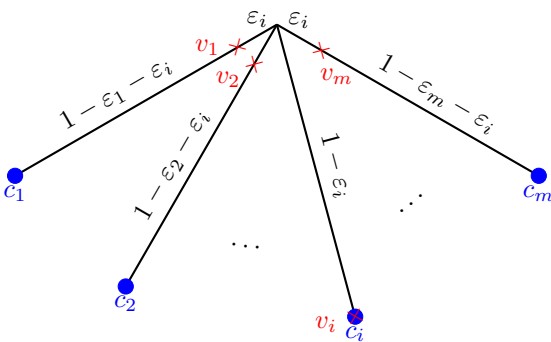

Figure 16: An illustration of the $i$-th instance on the star graph, where the $i$-th voter is located at the same spot as the $i$-th candidate. For each voter $v_j$, the closest candidate is $c_j$, followed by other candidates in descending order of $\varepsilon_k$. This ensures that all instances result in the same preference ordering for voters.

*Proof.* We proceed similarly to Theorem 16, using a family of instances $I_0, I_1, \ldots, I_m$ on the two-dimensional plane, such that the voters' ordinal preferences are the same in all of these instances, but the locations and distances of candidates and voters vary. In each instance, we have $m$ voters $\{v_1, v_2, \ldots, v_m\}$ and $m$ candidates $\{c_1, c_2, \ldots, c_m\}$.

Let $\ell = 3/\varepsilon - 1$. We use the same instance $I_0$ as in Theorem 16, where for each $i \in [m]$, candidate $c_i$ is located at $(-\ell, 2^{i-1}/2^m)$ and voter $v_i$ is located at $(0, 2^{i-1}/2^m)$. Then, for each $i \in [m]$, voter $v_i$ will have the ordinal preference

$$c_i \succ c_{i-1} \succ \cdots \succ c_1 \succ c_{i+1} \succ c_{i+2} \succ \cdots \succ c_m.$$

in instance $I_0$.

Next, we define instance $I_j$ for each $j \in [m]$. For each $i \in [m] \setminus \{j\}$, $v_i$ and $c_i$ will remain in the same location as in $I_0$. Meanwhile, candidate $c_j$ is moved to $(\ell, 2^{j-1}/2^m)$ and voter $v_j$ to $(2\ell, 2^{j-1}/2^m)$. It can be seen that for any $i \neq j$, the distance of voter $v_i$ to candidate $c_j$ will be unchanged compared to $I_0$, and therefore the voter's ordinal preference will remain the same. For voter $j$, its closest candidate will remain $c_j$. In addition, its preference for the other candidates will remain unchanged. Therefore, voter $v_j$'s ordinal preference will remain the same as in $I_0$, too, and all voters' ordinal preferences are identical in $I_0$ and $I_j$. An example of these instances is illustrated in Figure 17.

Now, consider an arbitrary deterministic algorithm ALG that selects $k < m$ candidates. Since $k < m$, when running ALG on $I_0$, there exists a candidate $c_j$ that is not in the set of selected candidates $C$. Now, consider the algorithm's performance on $I_j$. Since the voter's ordinal preferences are the same in $I_0$ and $I_j$, and ALG operates using only this information, its output on $I_j$ must be the same as in $I_0$ and therefore, it does not select $c_j$. Now, we can lower bound the cost of ALG with respect to the max objective as

$$\begin{aligned} \text{cost}_m(C, I_0) &\geq d(v_j, C) \\ &\geq 3\ell. \end{aligned} \qquad \text{(Since } c_j \notin C)$$

On the other hand, if we only select $c_j$, we get

$$\begin{aligned} \text{cost}_m(c_j, I_0) &= \max_{i \in [m]} \|v_i - c_j\|_2 \\ &\leq \ell + 1. \end{aligned}$$

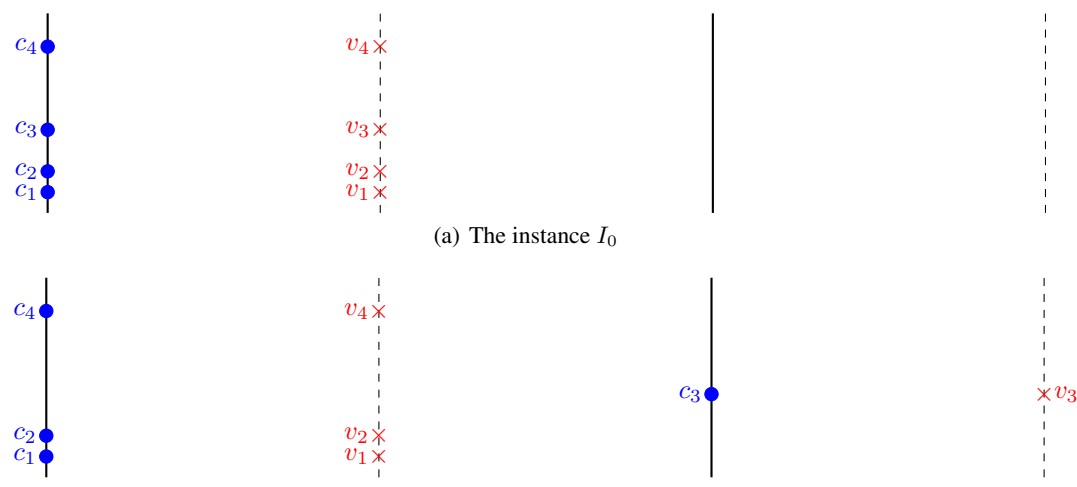

(a) The instance $I_0$

(b) $I_3$: Instance with voter $v_3$ and candidate $c_3$ moved

Figure 17: Figures of the lower bound instances for the max objective. In (a), all candidates are located on the line $x = -\ell$, with voters with matching $y$-coordinates on the line $x = 0$. In (b), candidate $c_3$ is moved to the line $x = \ell$, and voter $v_3$ is moved to the line $x = 2\ell$ while keeping the same $y$-coordinates.

This shows that the optimal cost $\mathrm{OPT}_m(I_j) \leq \ell + 1$. Finally, combining these two, we get that the distortion of ALG on instance $I_j$ is at least

$$
\begin{aligned}
\texttt{1-distortion(ALG)} &\geq \frac{\mathrm{cost}_m(C, I_k)}{\mathrm{OPT}_m(I_k)} \\
&\geq \frac{3\ell}{\ell + 1} \\
&= 3 - \frac{3}{\ell + 1} \\
&= 3 - \varepsilon. \qquad\qquad (\ell = 3/\varepsilon - 1)
\end{aligned}
$$

$\square$

**Theorem 19.** Any deterministic algorithm for the 2-committee election (with respect to the max-cost) when voters and candidates are located on a line must have a 1-distortion of at least $2 - \varepsilon$ for any $\varepsilon > 0$.

*Proof.* We consider the following instance with three voters $\{v_1, v_2, v_3\}$ and three candidates $\{c_1, c_2, c_3\}$. Let the preferences of voters be as follows:

$$
\begin{aligned}
v_1 &: c_1 \succ c_2 \succ c_3 \\
v_2 &: c_2 \succ c_1 \succ c_3 \\
v_3 &: c_3 \succ c_2 \succ c_1.
\end{aligned}
$$

Next, we consider three possible placements for the voters and candidates that respect the above (desired) preference order of voters, as shown in Figure 18. In all instances, $v_1$ is located at point $-1$, and $v_3$ at point $1$. In the first instance, $v_2$ is located at $0.5$. Candidates $c_1, c_2, c_3$ are located at points $0, 1 - \varepsilon$ and $1 + \varepsilon/2$ respectively. It is easy to see that the voters' preferences in this instance match our desired orders. Now, we can see that in this instance, $c_1$ has a distance of at most $1$ to every voter, while both $c_2$ and $c_3$ have a distance of at least $2 - \varepsilon$ from $v_1$. Therefore, not choosing $c_1$ will lead to a distortion of at least $2 - \varepsilon$.

In the second instance, we place $v_2$ and $c_2$ at point $0$, $c_1$ at point $-2 + \varepsilon$, and $c_3$ at point $2 - \varepsilon/2$. Again, we can see that the voters' preferences will follow the desired orders. In this instance, $c_2$

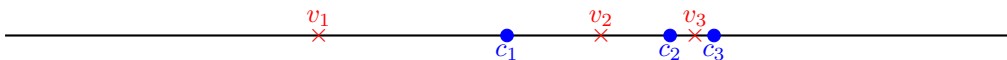

(a) First instance: Choosing $c_1$ achieves a cost of 1, while the other two candidates have a distances of at least $2 - \varepsilon$ to $v_1$.

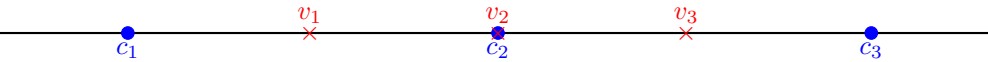

(b) Second instance: Choosing $c_2$ achieves a cost of 1, while the other two candidates have a distance of at least $2 - \varepsilon$ to $v_2$.

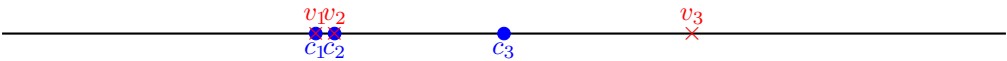

(c) Third instance: Choosing $c_3$ achieves a cost of 1, while the other two candidates have a distances of at least $2 - \varepsilon$ to $v_3$.

Figure 18: Possible locations of the voters and candidates in the lower bound instance for $k = 2$.

has a distance of at most 1 from all voters, while $v_2$ is at a distance of at least $2 - \varepsilon$ from the other candidates. Therefore, not choosing $c_2$ will lead to a distortion of at least $2 - \varepsilon$.

Finally, in the third instance, we place $c_1$ at point $-1$, $c_2$ and $v_2$ at point $-1 + \varepsilon$ and $c_3$ at point 0. This placement will also respect the desired preference orders for each voter. Additionally, since $c_3$ has a distance of at most 1 from all voters and $v_3$ has a distance of at least $2 - \varepsilon$ from the other candidates, not choosing $c_3$ leads to a distortion of at least $2 - \varepsilon$.

Now, since these instances cannot be distinguished based on the voters' ordinal preferences, and not choosing any of the candidates leads to a distortion of at least $2 - \varepsilon$, any deterministic algorithm selecting two candidates cannot achieve a distortion better than $2 - \varepsilon$.

$\square$

**Theorem 20.** Any deterministic algorithm for the 3-committee election (with respect to the max-cost) when voters and candidates are located on a line must have a 1-distortion of at least $3/2 - \varepsilon$ for any $\varepsilon > 0$.

*Proof.* We consider instances $\{I_1, I_2, I_3, I_4\}$, each with four voters $\{v_1, v_2, v_3, v_4\}$ and four candidates $\{c_1, c_2, c_3, c_4\}$ such that the preferences of voters are as follows in every instance:

$$v_1 : c_1 \succ c_2 \succ c_3 \succ c_4$$
$$v_2 : c_2 \succ c_1 \succ c_3 \succ c_4$$
$$v_3 : c_3 \succ c_4 \succ c_2 \succ c_1$$
$$v_4 : c_4 \succ c_3 \succ c_2 \succ c_1.$$

For the instance $I_i$, we choose the location of voters and candidates so that candidate $c_i$ has a distance of at most 2 to each voter, while voter $v_i$ has a distance of at least $3 - 2\varepsilon$ to every candidate except $c_i$. This leads to a distortion of at least $3/2 - \varepsilon$ for any deterministic algorithm ALG, as these instances cannot be distinguished based on ordinal preferences, and at least one candidate $c_i$ is not chosen in instance $I_i$ by ALG. These instances are shown in Figure 19.

In instance $I_1$, we have voters $v_1, v_2, v_3, v_4$ located at points $-2, 1 - \varepsilon, 2 - \varepsilon, 2$ and candidates $c_1, c_2, c_3, c_4$ located at points $0, 1, 2 - \varepsilon, 2$ respectively. It can be seen that the ordinal preference of each voter, in this instance, matches the desired ordering. Now, $c_1$ has a distance of at most 2 to every voter in this instance, while $v_1$ is at a distance of at least 3 from every candidate except $c_1$. So, if $c_1$ is not chosen, we get a distortion of at least $3/2$.

In instance $I_2$, we have voters $v_1, v_2, v_3, v_4$ located at points $-2, -1 - \varepsilon, 2 - \varepsilon, 2$ and candidates $c_1, c_2, c_3, c_4$ located at points $-4 + \varepsilon, 0, 2 - \varepsilon, 2$ respectively. Once again, we can see that the ordering

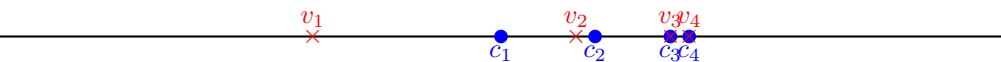

(a) $I_1$: Choosing $c_1$ achieves a cost of 2, while the other candidates have a distances of at least 3 to $v_1$.

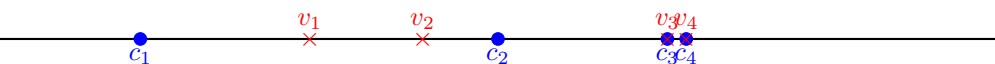

(b) $I_2$: Choosing $c_2$ achieves a cost of 2, while the other candidates have a distances of at least 3 to $v_2$.

Figure 19: Possible locations of the voters and candidates in the lower bound instance for $k = 3$.

for voters' preferences is respected:

$$d(v_1, c_1) = 2 - \varepsilon < d(v_1, c_2) = 2 < d(v_1, c_3) = 4 - \varepsilon < d(v_1, c_4) = 4$$
$$d(v_2, c_2) = 1 + \varepsilon < d(v_2, c_1) = 3 - 2\varepsilon < d(v_2, c_3) = 3 < d(v_1, c_4) = 3 + \varepsilon$$
$$d(v_3, c_3) = 0 < d(v_3, c_4) = \varepsilon < d(v_3, c_2) = 2 - \varepsilon < d(v_3, c_1) = 6 - 2\varepsilon$$
$$d(v_4, c_4) = 0 < d(v_4, c_3) = \varepsilon < d(v_4, c_2) = 2 < d(v_4, c_1) = 6 - \varepsilon.$$

In addition, since $c_2$ is at a distance of at most 2 from every voter and every candidate except $c_2$ has a distance of at least $3 - 2\varepsilon$ from $v_2$, not choosing $c_2$ leads to a distortion of at least $3/2 - \varepsilon$.

Based on the symmetry in voters' preferences between $c_1, c_2$ and $c_4, c_3$, we create mirror versions of $I_1$ and $I_2$ as $I_4$ and $I_3$ respectively, so that not choosing $c_4$ or $c_3$ would lead to a distortion of at least $3/2 - \varepsilon$. Therefore, since any deterministic algorithm must omit one of the candidates, we cannot achieve a distortion of better than $3/2 - \varepsilon$ in this case.

$\square$

