# OpenReview forum: "Bi-Criteria Metric Distortion"
_ICLR.cc/2026/Conference — ICLR 2026 Poster_

### Official Review · Reviewer_HLZ6 · 2025-10-23

**Soundness:** 3
**Presentation:** 3
**Contribution:** 2
**Rating:** 6
**Confidence:** 3

**Summary:**

This paper investigates the following problem in social choice theory under the metric distortion framework: if, instead of selecting a single winner, a deterministic mechanism is allowed to select multiple candidates (where the cost to a voter is their distance to the nearest selected candidate), then can this lead to smaller distortion relative to the optimal single candidate?

The authors obtained the following results:

1. Line metric (1D Euclidean): At most 4 candidates suffice to achieve optimal cost in both the utilitarian  and egalitarian cost setting.
2. 2-D Euclidean and tree metric: Even when selecting $m−1$ of $m$ candidates, the distortion cannot be smaller than $1+2/(m−1)$ for utilitarian and $3$ for egalitarian. As a corollary, it is impossible to achieve the optimal cost unless all candidates are selected.

**Strengths:**

This is a theoretical paper addressing a natural and interesting question in metric distortion. The results are logically sound and well contextualized within the existing literature. The writing is clear and well-organized.

**Weaknesses:**

1. The contribution is limited to deterministic algorithms and highly restricted metric classes (line, 2-D Euclidean, and tree metrics). It would be interesting to see whether similar results extend to randomized mechanisms or more general metrics.

2. The paper focuses on existence results without explicitly discussing the computational complexity or implementability of the proposed mechanisms.  Clarifying whether these mechanisms can be computed efficiently would strengthen the practical relevance of the work.

**Questions:**

See "weakness"

---

> ### Author Response · Authors · 2025-11-21
> **Response to Official Review by Reviewer HLZ6**
>
> Thank you for your positive assessment of the soundness and clarity, and for your comments on scope and implementability. Below we address your concerns and briefly indicate how the revised version reflects these.
>
> **Deterministic mechanisms and restricted metric classes.**
> We agree that considering other metric spaces and randomized mechanisms is important, but our focus on deterministic approaches in this paper is intentional. In the metric-distortion literature, deterministic and randomized methods form distinct lines of work: they achieve different guarantees, and their known lower bounds differ substantially.
>
> Moreover, our impossibility result (showing unbounded distortion for committees of size $\lceil \log(k+1) \rceil + 1$) also holds for randomized mechanisms, although we chose not to include that extension here given the paper’s deterministic scope.
>
> Finally, our negative results are not limited to 2D or tree metrics; rather, we show that these lower bounds hold even in these restricted metric spaces. This supports our focus on the line metric and demonstrates that achieving optimality in the bi-criteria setting is inherently difficult in slightly more general metric spaces.
>
> We agree, however, that extending the bi-criteria perspective to randomized mechanisms and to broader families of metric spaces would be highly interesting. In the revised version, we emphasize this direction explicitly in the conclusion.
>
> **Computational aspects and implementability.**
> Our mechanisms are fully constructive and rely on explicit algorithms that operate on the ordinal profile (for example, ordering candidates and voters on the line and selecting a small committee based on the resulting structure). Given this, it is straightforward to see that all our algorithms run in polynomial time. With mildly sophisticated data structures, one can even obtain near-linear running time in the input size $|V|\cdot|C|$ (up to polylogarithmic factors). However, such optimizations are beyond the scope of this work, as our focus is on achieving optimality guarantees within a reasonable polynomial runtime.
>
> We hope that our comments and revision address your concerns. If this is the case, we would be grateful if you would consider adjusting your score accordingly.

---

> > ### Comment · Reviewer_HLZ6 · 2025-11-26
> >
> > Thank you for the clarification. I encourage the authors to explicitly include these discussions in the revision, especially the extension of the impossibility result to randomized mechanisms and the statement of the near-linear time complexity.

---

### Official Review · Reviewer_rpfj · 2025-10-31

**Soundness:** 4
**Presentation:** 2
**Contribution:** 3
**Rating:** 4
**Confidence:** 2

**Summary:**

For reasons of convenience and practicality, when soliciting preferences we often see ordinal preferences. These preferences mask the true intensity of certain opinions and the gap between them - a small gap between A and B and a large gap between B and C are represented simply as A,B,C. And so the optimal solution can be ambiguous.

The authors propose an approach of committees' to address this well-established problem, in a bi-criteria sense: select a small k-committee but compare its cost against the best single candidate. They establish bounds on the maximum level of distortion this approach can have. While fairly simple intuitively, the authors prove their methods bounds and claims rigorously.

**Strengths:**

* I think the content of the paper is good, and I feel that between the appendices and the main body a strong paper can be found (more in weaknesses/suggestions)

* The content is well motivated, I believe it is a novel contribution, and well-justified.

**Weaknesses:**

* I thought the visuals in the paper were quite weak and to the detriment of the clarity of the paper. There are only 2 figures. The first is quite large but has conveys relatively little information. The next (2/3) one doesn't convey much beyond the writing. I felt that understandability could be easily enhanced with more visuals that convey more meaning (there are several in the appendix that seem to be much better/efficient at conveying information)

* Perhaps the above could be overcome, but the writing in section 3/4 was not, in my opinion, up to par. The lemmas are given in succession, the connections are weakly explained, and it's not presented in an easy way to follow. I recognize that there are proofs in the appendix and the space provided is limited, but the main text is not up to par. Between lemma 9-11 and then theorem 12 for instance, we cover 2 different cases in the span of a few sentences, with choppy flow.

**Questions:**

I think this paper is solid content-wise when the supplementary material is considered. I think the writing and presentation are not. I know much of the content is in the appendices, but as is, the writing in Sections 3 and 4 is not yet there in my opinion. That is my primary critique, and revision there could easily raise my score.

---

> ### Author Response · Authors · 2025-11-21
> **Response to Official Review by Reviewer rpfj**
>
> Thank you for your thoughtful evaluation and for pointing out specific issues with the presentation. Below we address your concerns and briefly indicate how the revised version reflects these.
>
> **Figures and visuals**
> Due to the page limit, we kept most figures in the appendix alongside the detailed proofs. We included a few figures in the main text to provide intuition, though we agree they were not as helpful as they could have been. In the revised version:
> * We reduced the size of the 2D example while retaining it, since it illustrates the general approach behind our counterexample showing that no constant-size committee can achieve the optimal solution.
> * We revised the figure for the max-cost objective to better illustrate why two candidates cannot be optimal while still achieving a 2-approximation.
>
>
> **Flow in Sections 3 and 4**
> You noted that the sequence of lemmas (e.g., around the max-cost upper-bound result) was hard to follow. In the revised version, we have added brief proof sketches and intuition for the key lemmas and theorems in the main text (highlighetd in blue) with full details remaining in the appendix.
>
> These revisions are intended specifically to address your concern that the main text was difficult to follow without the supplementary material. If they adequately resolve this issue, we would be grateful if you would consider updating your score.

---

> > ### Comment · Reviewer_rpfj · 2025-11-25
> > **Follow-up to authors**
> >
> > Thank you for your response, and apologies on my delayed response.
> >
> > I would appreciate further detail on what you've changed? I would appreciate slightly more detail on the layout. As I mentioned, I could be easily convinced to raise my score with some more detail here.
> >
> > Beyond this, I will monitor discussions with other reviewers, who  had more content related critiscms than I did.

---

> > > ### Comment · Reviewer_rpfj · 2025-11-26
> > > **Clarification**
> > >
> > > Apologies, I was looking at the original PDF and not the revision! I think this is a significant improvement. I will consider this; I am likely to increase my score but will think upon it.

---

### Official Review · Reviewer_xGkU · 2025-11-01

**Soundness:** 3
**Presentation:** 2
**Contribution:** 3
**Rating:** 6
**Confidence:** 3

**Summary:**

The paper investigates whether selecting multiple representatives (a committee) instead of a single candidate can reduce the efficiency loss that arises when elections rely only on ordinal preferences rather than full cardinal information. This problem is studied within the metric distortion framework, where voters and candidates are points in a metric space and the cost of selecting a candidate for a voter is their distance.

The authors introduce a bi-criteria approximation perspective, asking whether selecting a small committee of size k≥1 can yield a total or maximum voter cost close to that of an optimal single candidate. They show that in the line metric, it is indeed possible to achieve the optimal cost with only a constant number of candidates.

However, this improvement is shown to be unique to one-dimensional settings. In two-dimensional Euclidean and tree metrics, achieving optimal cost is impossible unless all candidates are selected.

**Strengths:**

The paper introduces a bi-criteria approximation framework to analyze how selecting multiple candidates can reduce efficiency loss in metric distortion, extending classical single-winner results to multi-candidate settings for both utilitarian and egalitarian objectives. It establishes precise trade-offs between committee size and approximation quality, proving that on the line metric two candidates can achieve optimal cost, while in higher-dimensional and tree metrics such improvement is impossible without selecting all candidates.

**Weaknesses:**

The main weakness of the paper is in the presentation, given that the introduction is too long and most proofs of correctness are relegated to the appendix, which is pretty long. On the other hand, the results are presented well and given that the paper is technically demanding, this choice is justified.

**Questions:**

No questions.

---

> ### Author Response · Authors · 2025-11-21
> **Response to Official Review by Reviewer xGkU**
>
> Thank you for your positive evaluation of the contribution and for your comments on presentation. Below we address your concerns and briefly indicate how the revised version reflects these.
>
> **Proofs in the appendix**
> You correctly observed that many proofs were only in the appendix. We thank the reviewer for this important observation. In the revised version, we have added brief proof sketches and the key intuition for the main lemmas and theorems directly in Sections 3 and 4 (highlighetd in blue), while keeping the full proofs in the appendices. This way, the core ideas are accessible in the main text and this concern is resolved.
>
> **Introduction length**
> We agree that the introduction might seem a bit lengthy but it was intended to convey the necessary intuition and context for our results. In the revised version, we streamlined several portions and added brief proof sketches, ensuring that the main body is self-contained while still clearly presenting all contributions.
>
> We hope that our comments and revision address your concerns. If this is the case, we would be grateful if you would consider adjusting your score accordingly.

---

### Official Review · Reviewer_VGd9 · 2025-11-01

**Soundness:** 4
**Presentation:** 3
**Contribution:** 3
**Rating:** 8
**Confidence:** 3

**Summary:**

The paper studies the question of whether it is possible to achieve the cost of an optimal candidate in the metric distortion framework when selecting only a fixed number of candidates instead of a single one. For the line metric, it provides a positive answer both for the utilitarian objective (with just 2 candidates) and the egalitarian objective (with just 4 candidates). It also provides matching lower bounds. In contrast, even in the 2D Euclidean metric, guaranteeing optimality requires selecting all candidates.

**Strengths:**

The paper studies a very natural question: how many additional candidates are needed to guarantee optimal distortion. This to some extend related with k-committee selection, but the main question posed here is different. The paper obtains strong and non-trivial results both both the utilitarian and egalitarian objectives, and both the 1D case and more general metrics. The results on the line are particularly surprising: a constant number of candidates (and in fact a small constant at that) always suffices to guarantee optimality. I find this result very interesting and informative in practice. In cases where one can potentially select more than one candidate, the results of this paper explain how one can navigate the tradeoff between picking a small number of candidates and guaranteeing small distortion. The paper also provides an interesting impossibility result, showing that a similar guarantee cannot be attained in general metrics. In fact, selecting all but one candidates will always fail to guarantee optimality. This provides an interesting separation between 1D and more general metrics, something which hasn't been investigated a lot in this line of work, with some exceptions. Overall, the paper makes a concrete contribution to a natural problem. It provides a comprehensive understanding under different objectives and metric spaces, together with matching lower bounds.

Furthermore, the writing of the paper is very good. It accurately places its contributions in the context of the existing work; all relevant references have been discussed. The technical component is also non-trivial and explained well in the main body.

**Weaknesses:**

The main weakness is that, a priori, selecting more than one winning candidate could trivialize the problem when comparing against the optimal omniscient solution that picks a single candidate. Still, the lower bounds of the paper show that this is far from trivial in general, and accomplishing it in 1D turns out to be highly non-trivial.

**Questions:**

- Is there some general property of the metric space that could be used to parameterize the number of candidates needed to reach optimality? I am trying to understand whether it is really only the 1D metric that has this nice property. Could you prove that, in some sense, any metric that is not isomorphic to the 1D will always fail to reach optimality with all but one candidates?

---

> ### Author Response · Authors · 2025-11-21
> **Response to Official Review by Reviewer VGd9**
>
> Thank you for the very positive assessment and for highlighting the interest of the 1D results and the separation from higher-dimensional metrics. Below we address your concerns and briefly indicate how the revised version reflects these.
>
>
> **On comparing to the single-winner optimum**
> As you mentioned, our results demonstrate that allowing a constant-size committee does not trivialize the comparison to the optimal single candidate. Even on the line, proving that a committee of size 2 (sum-cost) or 4 (max-cost) achieves 1-distortion 1 requires non-trivial structural arguments, and our lower bound results in 2D and on trees show that even very large committees (up to size $m-1$) cannot guarantee optimal cost in more general metrics.
>
>
> **On characterizing metric spaces where small committees achieve optimality**
> Our results currently establish a separation rather than a full characterization. On the line metric, we show that a constant-size committee can always match the optimal single candidate. In contrast, for both 2D Euclidean and tree metrics, even a committee of size $(m-1)$ cannot guarantee optimal cost relative to the best single candidate under the objectives we consider. Fully characterizing the metric spaces in which constant-size committees suffice remains an open problem, and we now state this explicitly as a direction for future work in the conclusion.
>
>
> We hope that our comments and revision address your concerns.

---

> > ### Comment · Reviewer_VGd9 · 2025-11-25
> >
> > I thank the authors for their response. I maintain my positive evaluation.

---

### Author Response · Authors · 2025-12-03

We sincerely thank the reviewers for their thoughtful and constructive feedback, as well as the area chair for overseeing the process. We carefully addressed all technical and presentation concerns raised in the reviews, and we appreciate the reviewers’ follow-up comments indicating that these issues have now been resolved. In particular, we note and are grateful for the indication that one reviewer was considering a higher score based on the revisions.

In addition, we will explicitly incorporate the clarification regarding complexity and the result on randomized mechanisms in the camera-ready version to further strengthen the presentation.

The revised manuscript is now technically solid and clearly presented. We respectfully believe that our results provide a meaningful contribution to the study of metric distortion and social choice, and we thank you for your careful consideration.

---

### Meta-Review · Area_Chair_aday · 2026-01-05

**Summary:**

1. Clarity issues in writing, e.g., the intro is long, the figures are not clear enough
2. The results focus on a deterministic setting, instead ofa  randomized setting

**Reviewer Concerns:**

All concern have been well addressed

**Reviewer Scores:**

Reviewer rpfj may increase its score from 4 to 6.

---

### Decision · Program_Chairs · 2026-01-26

Accept (Poster)